# Unsupervised Domain Adaptation by Learning Using Privileged Information

**Adam Breitholtz**\*  
*Department of Computer Science*  
*Chalmers University of Technology*
*adambre@chalmers.se*

**Anton Matsson**\*  
*Department of Computer Science*  
*Chalmers University of Technology*
*antmats@chalmers.se*

**Fredrik D. Johansson**  
*Department of Computer Science*  
*Chalmers University of Technology*
*fredrik.johansson@chalmers.se*

**Reviewed on OpenReview:** *https://openreview.net/forum?id=saV3MPH0kw*

## Abstract

Successful unsupervised domain adaptation is guaranteed only under strong assumptions such as covariate shift and overlap between input domains. The latter is often violated in high-dimensional applications like image classification which, despite this limitation, continues to serve as inspiration and benchmark for algorithm development. In this work, we show that training-time access to side information in the form of auxiliary variables can help relax restrictions on input variables and increase the sample efficiency of learning at the cost of collecting a richer variable set. As this information is assumed available only during training, not in deployment, we call this problem unsupervised domain adaptation by learning using privileged information (DALUPI). To solve this problem, we propose a simple two-stage learning algorithm, inspired by our analysis of the expected error in the target domain, and a practical end-to-end variant for image classification. We propose three evaluation tasks based on classification of entities in photos and anomalies in medical images with different types of available privileged information (binary attributes and single or multiple regions of interest). We demonstrate across these tasks that using privileged information in learning can reduce errors in domain transfer compared to baselines, be robust to spurious correlations in the source domain, and increase sample efficiency.

## 1 Introduction

Deployment of machine learning (ML) systems relies on generalization from training samples to new instances in a target domain. When these new instances differ in distribution from the source of training data, performance tends to degrade and guarantees are often weak. For example, a supervised ML model trained to identify medical conditions in X-ray images from one hospital may work poorly in another hospital if the two sites have different equipment or examination protocols (Zech et al., 2018). In the *unsupervised domain adaptation* (UDA) problem (Ben-David et al., 2006), *no* labeled examples are available from the target domain and strong assumptions are needed for success. In this work, we ask: How can access to *auxiliary variables* during training help solve the UDA problem and weaken the assumptions necessary to guarantee domain transfer?

In standard UDA, a common assumption is that the object of the learning task is identical in source and target domains but that input distributions differ (Shimodaira, 2000). This "covariate shift" assumption is

---

\*Equal contribution.

plausible in our X-ray example above: Doctors are likely to give the same diagnosis based on X-rays of the same patient from similar but different equipment. However, guarantees for consistent domain adaptation also require either distributional overlap between inputs from source and target domains or known parametric forms of the labeling function (Ben-David & Urner, 2012; Wu et al., 2019; Johansson et al., 2019). Without these, adaptation cannot be verified or guaranteed by statistical means.

Input domain overlap is implausible for the high-dimensional tasks that have become standard benchmarks in the UDA community, including image classification (Long et al., 2013; Ganin et al., 2016) and sentence labeling (Orihashi et al., 2020). If hospitals have different X-ray equipment, the probability of observing (near-)identical images from source and target domains is zero (Zech et al., 2018). Even when covariate shift and overlap are satisfied, large domain differences can have a dramatic effect on sample complexity (Breitholtz & Johansson, 2022). Despite promising developments (Shen et al., 2022), realistic guarantees for practical domain transfer remain elusive.

In supervised ML without domain shift, incorporating auxiliary variables in the training of models has been proposed to improve out-of-sample generalization. For example, learning using *privileged information* (Vapnik & Vashist, 2009; Lopez-Paz et al., 2016), variables available during training but unavailable in deployment, has been proven to require fewer examples compared to learning without these variables (Karlsson et al., 2021). In X-ray classification, privileged information (PI) can come from graphical annotations or clinical notes made by radiologists that are unavailable when the system is used. While PI has begun to see use in domain adaptation, see e.g., Sarafianos et al. (2017) or Vu et al. (2019), and a theoretical analysis exists for linear classifiers (Xie et al., 2020), the literature has yet to fully characterize the benefits of this practice.

We introduce *unsupervised domain adaptation by learning using privileged information* (DALUPI), in which auxiliary variables, related to the outcome of interest, are leveraged during training to improve test-time adaptation when the variables are unavailable. We summarize our contributions below:

- We formalize the DALUPI problem and give conditions under which it is possible to solve it consistently, i.e., to learn a model using privileged information that predicts optimally in the target domain. Importantly, these conditions do not rely on distributional overlap between source and target domains in the input variable (Section 2.1), making consistent learning without privileged information (PI) generally infeasible.

- We propose practical learning algorithms for image classification in the DALUPI setting (Section 3), designed to handle problems with three different types of PI, see Figure 1 for examples. As common UDA benchmarks lack auxiliary variables related to the learning problem, we propose three new evaluation tasks spanning the three types of PI using data sets with real-world images and auxiliary variables.

- On these tasks, we compare our methods to supervised learning baselines and well-known methods for unsupervised domain adaptation (Section 4). We find that our proposed models perform favorably to the alternatives for all types of PI, particularly when input overlap is violated and when training sets are small.

## 2 Privileged Information in Domain Adaptation

In unsupervised domain adaptation (UDA), the goal is to learn a hypothesis $h$ to predict outcomes (or labels) $Y \in \mathcal{Y}$ for problem instances represented by input covariates $X \in \mathcal{X}$, drawn from a target domain with density $\mathcal{T}(X, Y)$. During training, we have access to labeled samples $(x, y)$ only from a source domain $\mathcal{S}(X, Y)$ and unlabeled samples $\tilde{x}$ from $\mathcal{T}(X)$. As a running example, we think of $\mathcal{S}$ and $\mathcal{T}$ as radiology departments at two different hospitals, of $X$ as the X-ray image of a patient, and of $Y$ as the diagnosis made by a radiologist after analyzing the image.

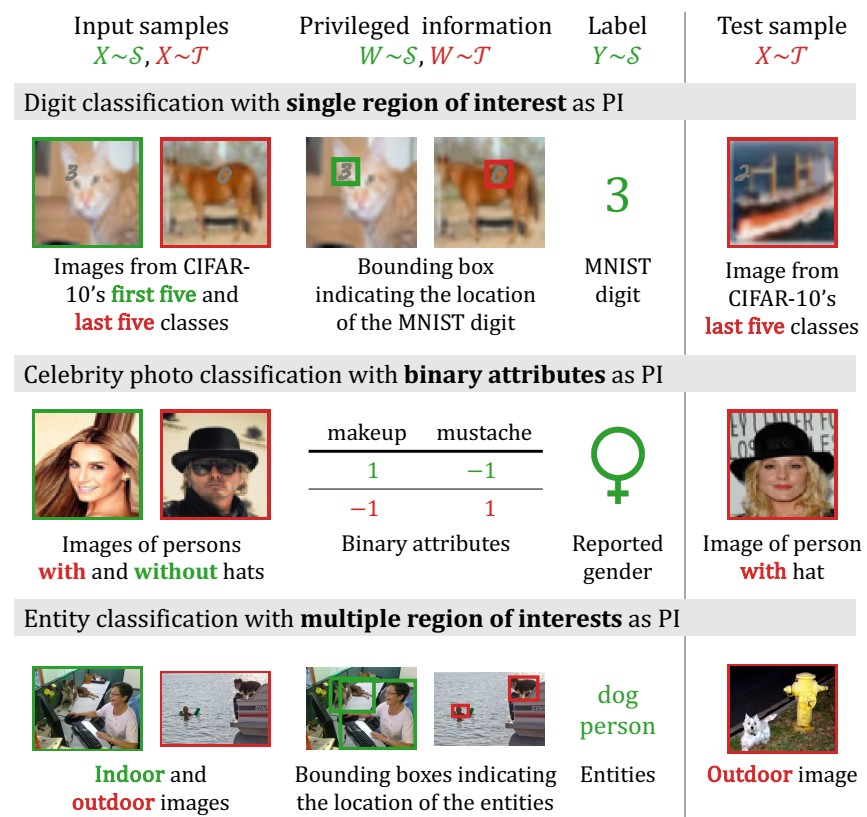

Figure 1: Examples of domain adaptation tasks with different types of privileged information (PI). During training, input samples $X$ and PI $W$ are drawn from both source and target domains. Labels $Y$ are only available from the source domain. At test time, a target sample $X$ is observed. We consider three types of PI: binary attribute vectors, a single region of interest, and multiple regions of interest.

We aim to learn a hypothesis $h \in \mathcal{H}$ from a hypothesis set $\mathcal{H}$ that minimizes the expected target-domain prediction error (risk) $R_{\mathcal{T}}$, with respect to a loss function $L : \mathcal{Y} \times \mathcal{Y} \to \mathbb{R}_+$, i.e., to solve

$$\min_{h \in \mathcal{H}} R_{\mathcal{T}}(h), \quad R_{\mathcal{T}}(h) \coloneqq \mathbb{E}_{\mathcal{T}(X,Y)}[L(h(X), Y)] , \tag{1}$$

where we use the subscript convention $\mathbb{E}_{p(X)}[f(X)] = \int_{x \in \mathcal{X}} p(x)f(x)dx$ to denote an expectation of some function $f$ over a density $p$ on the domain $\mathcal{X}$. A consistent solution to the UDA problem returns a minimizer of Equation 1 without ever observing labeled samples from $\mathcal{T}$. However, if $\mathcal{S}$ and $\mathcal{T}$ are allowed to differ arbitrarily, finding such a solution cannot be guaranteed (Ben-David & Urner, 2012). To make the problem feasible, we assume that *covariate shift* (Shimodaira, 2000) holds—that the labeling function is the same in both domains, but the covariate distributions differ.

**Assumption 1** (Covariate shift). *For domains $\mathcal{S}, \mathcal{T}$ on $\mathcal{X} \times \mathcal{Y}$, covariate shift holds with respect to $X$ if*

$$\exists x \in \mathcal{X} : \mathcal{T}(X = x) \neq \mathcal{S}(X = x) \text{ and } \forall x \in \mathcal{X} : \mathcal{T}(Y \mid x) = \mathcal{S}(Y \mid x) .$$

In our example, covariate shift means that radiologists at either hospital would diagnose two patients with the same X-ray in the same way, but that the radiologists may encounter different distributions of patients and images. To guarantee consistent learning without further assumptions, these distributions cannot be *too* different—the source input domain $\mathcal{S}(x)$ must sufficiently *overlap* the target input domain $\mathcal{T}(x)$.

**Assumption 2** (Domain overlap). *A domain $\mathcal{S}$ overlaps another domain $\mathcal{T}$ with respect to $X$ on $\mathcal{X}$ if*

$$\forall x \in \mathcal{X} : \mathcal{T}(X = x) > 0 \implies \mathcal{S}(X = x) > 0 .$$

Table 1: A summary of the different settings we consider in this work, what data is assumed to be available during training and if guarantees for identification are known for the setting under the assumptions of Proposition 1. The parentheses around source samples for DALUPI indicate that we need not necessarily observe these for the setting. Note that at test time only $x$ from $\mathcal{T}$ is observed. *Under the more generous assumption of overlapping support in the input space $\mathcal{X}$, guarantees exist for all these settings.

| Setting | Observed $\mathcal{S}$ | | | Observed $\mathcal{T}$ | | | Guarantee |
| | $x$ | $w$ | $y$ | $\tilde{x}$ | $\tilde{w}$ | $\tilde{y}$ | for $R_{\mathcal{T}}$ |
|---|---|---|---|---|---|---|---|
| SL-T | | | | ✓ | | ✓ | ✓ |
| SL-S | ✓ | | ✓ | | | | * |
| UDA | ✓ | | ✓ | ✓ | | | * |
| LUPI | ✓ | ✓ | ✓ | | | | * |
| DALUPI | (✓) | ✓ | ✓ | ✓ | ✓ | | ✓ |

Covariate shift and domain overlap with respect to $X$ guarantee that the target risk $R_{\mathcal{T}}$ can be identified by the sampling distribution described above, and thus, that a solution to Equation 1 may be found. Hence, they have become standard assumptions, used by most informative guarantees (Zhao et al., 2019).

Overlap is often violated in high-dimensional problems such as image classification, partly due to irrelevant information that has a spurious association with the label $Y$ (Beery et al., 2018; D'Amour et al., 2021). In X-ray classification, it may be possible to perfectly distinguish hospitals (domains) based on protocol or equipment differences manifesting in the images (Zech et al., 2018). There are no guarantees for optimal UDA in this case. Some guarantees based on distributional distances do not rely on overlap (Ben-David et al., 2006; Long et al., 2013), but do not guarantee optimal learning either (Johansson et al., 2019).

Still, an image $X$ may *contain* information $W$ which is both *sufficient for prediction* and *supported in both domains*. For X-rays, this could be a region of pixels indicating a medical condition, ignoring parts that merely indicate differences in protocol (Zech et al., 2018). The learner does not know how to find this information a priori, but it can be supplied during training as added supervision. A radiologist could indicate regions of interest $W$ using bounding boxes during training (Irvin et al., 2019), but would not be available to annotate images at test time. As such, $W$ is *privileged information* (Vapnik & Vashist, 2009).

## 2.1 Unsupervised Domain Adaptation With Privileged Information

Learning using privileged information, variables that are available only during training but not at test time, has been shown to improve sample efficiency in diverse settings (Vapnik & Izmailov, 2015; Pechyony & Vapnik, 2010; Jung & Johansson, 2022). Next, we show that privileged information can also improve UDA by providing *identifiability* of the target risk—allowing it to be computed from the sampling distribution—even when overlap is not satisfied in $X$.

We define domain adaptation by learning using privileged information (DALUPI) as follows. During training, learners observe samples of covariates $X$, labels $Y$ and privileged information $W \in \mathcal{W}$ from $\mathcal{S}$ in a dataset $D_{\mathcal{S}} = \{(x_i, w_i, y_i)\}_{i=1}^{m}$, as well as samples of covariates and privileged information from $\mathcal{T}$, $D_{\mathcal{T}} = \{(\tilde{x}_i, \tilde{w}_i)\}_{i=1}^{n}$. *At test time, trained models only observe covariates $\tilde{x} \sim \mathcal{T}(X)$ and our learning goal remains to minimize the target risk, see Equation 1.* We justify access to privileged information from $\mathcal{T}$, but not labels, by pointing out that it is often easier to annotate observations with privileged information $W$ than with labels $Y$. For example, a non-expert may be able to reliably recognize the outline of an animal in an image, indicating the pixels $W$ corresponding to it, but not identify its species ($Y$); see Figure 2, where it would likely be easier to identify the location of the cat in the image than to identify its breed.

To identify $R_{\mathcal{T}}$ (Equation 1) without overlap in $X$, we make the assumption that $W$ is sufficient to predict $Y$ in the following sense.

**Assumption 3** (Sufficiency of privileged information). *Privileged information $W$ is sufficient for the outcome $Y$ given covariates $X$ if $Y \perp X \mid W$ in both $\mathcal{S}$ and $\mathcal{T}$.*

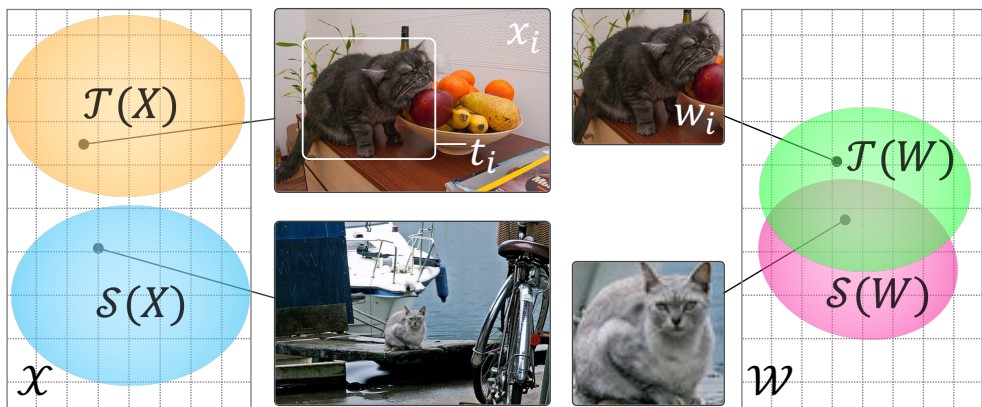

Figure 2: An illustration of domain overlap being more plausible when we consider appropriate forms of privileged information $W$, such as a region of interest of an image. Source and target domains $\mathcal{S}, \mathcal{T}$ are here indoor and outdoor images $X$ and the task is to identify the animal $Y$ in the image.

Assumption 3 is satisfied when $X$ provides no more information about $Y$ in the presence of $W$. If we consider $W$ to be a subset of image pixels corresponding to an area of interest, the other pixels in $X$ may be unnecessary to predict $Y$. This is illustrated in Figure 2 where the privileged information $w_i$ is the region of interest indicated by the bounding box $t_i$. Here, overlap is more probable in $\mathcal{W}$ than in $\mathcal{X}$, as the extracted pixels mostly show cats. Moreover, when $W$ retains more information, sufficiency becomes more plausible but domain overlap in $W$ is reduced. The sufficiency assumption is used to replace $\mathcal{T}(y \mid x)$ with $\mathcal{T}(y \mid w)$ in Proposition 1. If sufficiency is violated but it is plausible that the degree of insufficiency is comparable across domains, we can still obtain a bound on the target risk which may be estimated from observed quantities. We give such a result in Appendix F.

We expect that some PI can be selected to be sufficient for a given task. However, if this sufficiency cannot be ensured, the overall performance may decrease, assuming covariate shift with respect to $W$ is not violated. Even so, we still anticipate the generalization error to remain of a comparable magnitude. If covariate shift is violated in $W$, further performance declines are expected, as the problem becomes more complex and we are not guaranteed to identify the optimal hypothesis (Johansson et al., 2019).

Assumptions 1–2 holding with respect to privileged information $W$ instead of $X$, along with Assumption 3, allow us to identify the target risk even for models $h \in \mathcal{H}$ that do not use $W$ as input:

**Proposition 1.** *Let Assumptions 1 and 2 be satisfied with respect to $W$ (not necessarily with respect to $X$) and let Assumption 3 hold as stated. Then, the target risk $R_\mathcal{T}$ is identified for hypotheses $h : \mathcal{X} \to \mathcal{Y}$,*

$$R_\mathcal{T}(h) = \mathbb{E}_{\mathcal{T}(X)} \left[ \mathbb{E}_{\mathcal{T}(W|X)} \left[ \mathbb{E}_{\mathcal{S}(Y|W)}[L(h(X), Y) \mid X, W] \mid X \right] \right]$$
$$= \int_x \mathcal{T}(x) \int_w \mathcal{T}(w \mid x) \int_y \mathcal{S}(y \mid w) L(h(x), y) \mathrm{d}y \mathrm{d}w \mathrm{d}x \ ,$$

*and for $L$ the squared loss, a minimizer of $R_\mathcal{T}$ is the function*

$$h_\mathcal{T}^*(x) = \mathbb{E}_{\mathcal{T}(W|x)}[\mathbb{E}_{\mathcal{S}(Y|W)}[Y \mid W] \mid x] = \int_w \mathcal{T}(w \mid x) \int_y \mathcal{S}(y \mid w) y \ \mathrm{d}y \mathrm{d}w \ .$$

*Proof sketch.* $R_\mathcal{T}(h) = \int_{x,y} \mathcal{T}(x, y) L(h(x), y) \mathrm{d}x \mathrm{d}y$. We can then marginalize over $W$ to get $\mathcal{T}(x, y) = \mathcal{T}(x) \mathbb{E}_{\mathcal{T}(W|x)}[\mathcal{T}(y \mid W) \mid x] = \mathcal{T}(x) \int_{w:\mathcal{S}(w)>0} \mathcal{T}(w \mid x) \mathcal{S}(y \mid w) \mathrm{d}w$, where the first equality follows by sufficiency and the second by covariate shift and overlap in $W$. $\mathcal{T}(x), \mathcal{T}(w \mid x)$ and $\mathcal{S}(y \mid w)$ are observable through training samples. That $h_\mathcal{T}^*$ is a minimizer follows from the first-order condition. See Appendix C. □

Proposition 1 shows that there are conditions where privileged information allows for identification of target-optimal hypotheses where identification is not possible without it, i.e., when overlap is violated in $X$. $W$

guides the learner toward the information in $X$ that is relevant for the label $Y$. When $W$ is deterministic in $X$, overlap in $X$ implies overlap in $W$, but not vice versa. In the same case, under Assumption 3, if covariate shift holds for $X$, it holds also for $W$. Hence, if sufficiency can be justified, the requirements on $X$ are weaker than in standard UDA, at the cost of collecting $W$. Surprisingly, Proposition 1 does not require that $X$ is observed in the source domain as the result does not depend on $\mathcal{S}(x)$.

Figure 1 gives examples of problems with the DALUPI structure which we consider in this work. For comparison, we list related learning paradigms in Table 1. Supervised learning (SL-S) refers to learning from labeled samples from $\mathcal{S}$ without privileged information. SL-T refers to supervised learning with (infeasible) access to labeled samples from $\mathcal{T}$. UDA refers to the setting at the start of Section 2 and LUPI to learning using privileged information without data from $\mathcal{T}$ (Vapnik & Vashist, 2009). We compare DALUPI to these alternative settings in our experiments in Section 4.

## 2.2 A Two-stage Algorithm and Its Risk

In light of Proposition 1, a natural learning strategy is to model privileged information as a function of the input, $\mathcal{T}(W \mid x)$, and the outcome as a function of privileged information, $\hat{g}(w) \approx \mathbb{E}_{\mathcal{S}}[Y \mid w]$, and combining these. In the case where $W$ is a deterministic function of $X$, $\mathcal{T}(W \mid x)$ is a map $f : \mathcal{X} \to \mathcal{W}$, which may be estimated as a regression $\hat{f}$ and combined with the outcome regression to form $\hat{h} = \hat{g}(\hat{f}(X))$. We may find such functions $\hat{f}, \hat{g}$ by separately minimizing the empirical risks

$$\hat{R}_{\mathcal{T}}^W(f) = \frac{1}{n} \sum_{i=1}^n L_W(f(\tilde{x}_i), \tilde{w}_i) \quad \text{and} \quad \hat{R}_{\mathcal{S}}^Y(g) = \frac{1}{m} \sum_{i=1}^m L_Y(g(w_i), y_i) . \tag{2}$$

Hypothesis classes $\mathcal{F}, \mathcal{G}$ may be chosen so that $\mathcal{H} = \{h = g \circ f; (f, g) \in \mathcal{F} \times \mathcal{G}\}$ has a desired form. Note that $L_W$ and $L_Y$ may in general be different loss functions.

We can bound the generalization error of estimators $\hat{h} = \hat{g} \circ \hat{f}$ when $W \in \mathbb{R}^{d_W}$ and the loss is the squared loss. We do this by placing an assumption of Lipschitz smoothness on the space of prediction functions: $\forall g \in \mathcal{G}, w, w' \in \mathcal{W} : \|g(w) - g(w')\|_2 \leq M\|w - w'\|_2$. To arrive at a bound, we first define the $\rho$-weighted empirical risk of the outcome model $g$ in the source domain, $\hat{R}_{\mathcal{S}}^{Y, \rho}(g) = \frac{1}{m} \sum_{i=1}^m \rho(w_i) L\_W(g(w_i), y_i)$ where $\rho$ is the density ratio of $\mathcal{T}$ and $\mathcal{S}$, $\rho(w) = \frac{\mathcal{T}(w)}{\mathcal{S}(w)}$. When the density ratio $\rho$ is unknown, we may use density estimation (Sugiyama et al., 2012) or probabilistic classifiers to estimate it. We arrive at the following result, proven for univariate $Y$ but generalizable to multivariate outcomes.

**Proposition 2.** *Suppose that $W$ and $Y$ are deterministic in $X$ and $W$, respectively, and that Assumptions 1–3 hold with respect to $W$. Let $\mathcal{G}$ comprise $M$-Lipschitz mappings $g : \mathcal{W} \to \mathcal{Y}$ with $\mathcal{W} \subseteq \mathbb{R}^{d_W}$, and let the loss be the squared Euclidean distance, assumed to be uniformly bounded over $\mathcal{W}$. Let $\rho(w) = \mathcal{T}(w)/\mathcal{S}(w)$ and $d$ and $d'$ be the pseudo-dimensions of $\mathcal{G}$ and $\mathcal{F}$, respectively. Assume that there are $m$ labeled samples from $\mathcal{S}$ and $n$ unlabeled samples from $\mathcal{T}$. Then, for any $h = g \circ f \in \mathcal{G} \times \mathcal{F}$, with probability at least $1 - \delta$,*

$$\frac{R_{\mathcal{T}}(h)}{2} \leq \hat{R}_{\mathcal{S}}^{Y, \rho}(g) + M^2 \hat{R}_{\mathcal{T}}^W(f) + \mathcal{O}\left( \sqrt[3/8]{\frac{d \log \frac{m}{d} + \log \frac{4}{\delta}}{m}} + \sqrt{\frac{d' \log \frac{n}{d'} + \log \frac{d_W}{\delta}}{n}} \right) .$$

*Proof sketch.* Decomposing the risk of $h \circ \phi$ , we get

$$
\begin{aligned}
R_{\mathcal{T}}(h) &= \mathbb{E}_{\mathcal{T}}[(g(f(X)) - Y)^2] \\
&\leq 2\mathbb{E}_{\mathcal{T}}[(g(W) - Y)^2 + (g(f(X)) - g(W))^2] \\
&\leq 2\mathbb{E}_{\mathcal{T}}[(g(W) - Y)^2 + M^2 \|f(X)) - g(W)\|^2] \\
&\leq 2\mathbb{E}_{\mathcal{T}}[(g(W) - Y)^2] + 2M^2 \mathbb{E}_{\mathcal{T}}[\|(f(X) - W)\|^2] \\
&= 2R_{\mathcal{T}}^Y(g) + 2M^2 R_{\mathcal{T}}^W(f) = 2R_{\mathcal{S}}^{Y, \rho}(g) + 2M^2 R_{\mathcal{T}}^W(f) .
\end{aligned}
$$

The first inequality follows the relaxed triangle inequality, the second from the Lipschitz property, and the third equality from Overlap and Covariate shift. Treating each component of $\hat{w}$ as independent, using

standard PAC learning results, and application of Theorem 3 from Cortes et al. (2010) allows us to reweight the risk with the density ratio $\rho$ by also adding an additional term which contains the Rényi divergence. Then with a union bound argument, we get the stated result. See Appendix D for a more detailed proof. $\quad\square$

When $\mathcal{F}$ and $\mathcal{G}$ contain the ground-truth mappings between $X$ and $W$ and between $W$ and $Y$, in the infinite-sample limit, minimizers of Equation 2 minimize $R_\mathcal{T}$ as well. Our approach is not limited to classical PAC analysis but could, under suitable assumptions, be carried out under another framework, e.g. using PAC-Bayes analysis to obtain a bound that contains different sample complexity terms. However, such a bound would then hold in expectation over a posterior distribution on $\mathcal{H}$ instead of uniformly over $\mathcal{H}$. We sketch a proof of such a bound in Appendix E.

Furthermore, if sufficiency is violated but it is plausible that the degree of insufficiency is comparable across domains, we can still obtain a bound on the target risk which may be estimated from observed quantities. We give such a result in Appendix F.

## 3 Image Classification With Privileged Information

We use image classification, where $X$ is an image and $Y$ is a discrete label, as proof of concept for DALUPI. To show the versatility of our approach, we consider three different instantiations of privileged information $W$: a binary attribute vector, a single region of interest, or multiple regions of interest. The two-stage estimator, see Figure 3a, is used in the first two cases. With multiple regions of interest as privileged information, we use an end-to-end model based on Faster-R-CNN (Ren et al., 2016), see Figure 3b. We detail each setting below and illustrate them in Figure 1.

### 3.1 Binary Attributes as PI

First, we consider the case where each image $x_i$ is accompanied by privileged information in the form of a binary vector $w_i \in \{0,1\}^d$ indicating the presence of $d$ attributes in the image. In this setting, we can directly apply our two-stage estimator (Equation 2). For the first estimator $\hat{f}$, we use a convolutional neural network (CNN) trained on observations from $\mathcal{T}$ (and possibly $\mathcal{S}$) to output a vector of attributes $\hat{w}_i$ from the input $x_i$. For the second estimator $\hat{g}$, we use a multi-layer perceptron classifier, trained on source domain observations, that predicts the image label $\hat{y}_i$ given the vector of attributes $w_i$. We use the categorical cross-entropy loss to train both $\hat{f}$ and $\hat{g}$. The resulting classifier, $\hat{h}(x) = \hat{g}(\hat{f}(x))$, is subsequently evaluated on target domain images.

### 3.2 Single Region of Interest as PI

Next, we consider privileged information as a subset of pixels $w_i$, taken from the image $x_i$ and associated with an object or feature that determines the label $y_i \in \{1, \ldots, K\}$. In our experiments, this PI is provided as a *single* bounding box with coordinates $t_i \in \mathbb{R}^4$ enclosing the region of interest $w_i$. Here, we use two CNNs, $\hat{d}$ and $\hat{g}$, and a deterministic function $\phi$ to approximate the two-stage estimator (Equation 2). The network $\hat{d}$ is trained to output bounding box coordinates $\hat{t}_i$ as a function of the input $x_i$, and the pixels $\hat{w}_i$ within the bounding box are extracted from $x_i$ and resized to pre-specified dimensions through $\phi$. The composition of these two functions, $\hat{f}(x_i) = \phi(x_i, \hat{d}(x_i))$, returns $\hat{w}_i$. The second network $\hat{g}$ is trained to predict $y_i$ given the pixels $w_i$ contained in a bounding box $t_i$ based on observations from $\mathcal{S}$. We use the mean squared error loss for $\hat{d}$ and the categorical cross-entropy loss for $\hat{g}$. Finally, $\hat{h}(x) = \hat{g}(\hat{f}(x))$ is evaluated on target domain images where the output of $\hat{f}$ is used for prediction with $\hat{g}$. See Appendix A.1 for further details.

### 3.3 Multiple Regions of Interest as PI

Finally, we consider a setting where privileged information indicates *multiple* regions of interest in an image. We use this PI in multi-label classification problems where the image $x_i$ is associated with one or more categories $k$ from a set $\{1, \ldots, K\}$, encoded in a multi-category label $y_i \in \{0,1\}^K$ (e.g., indicating findings

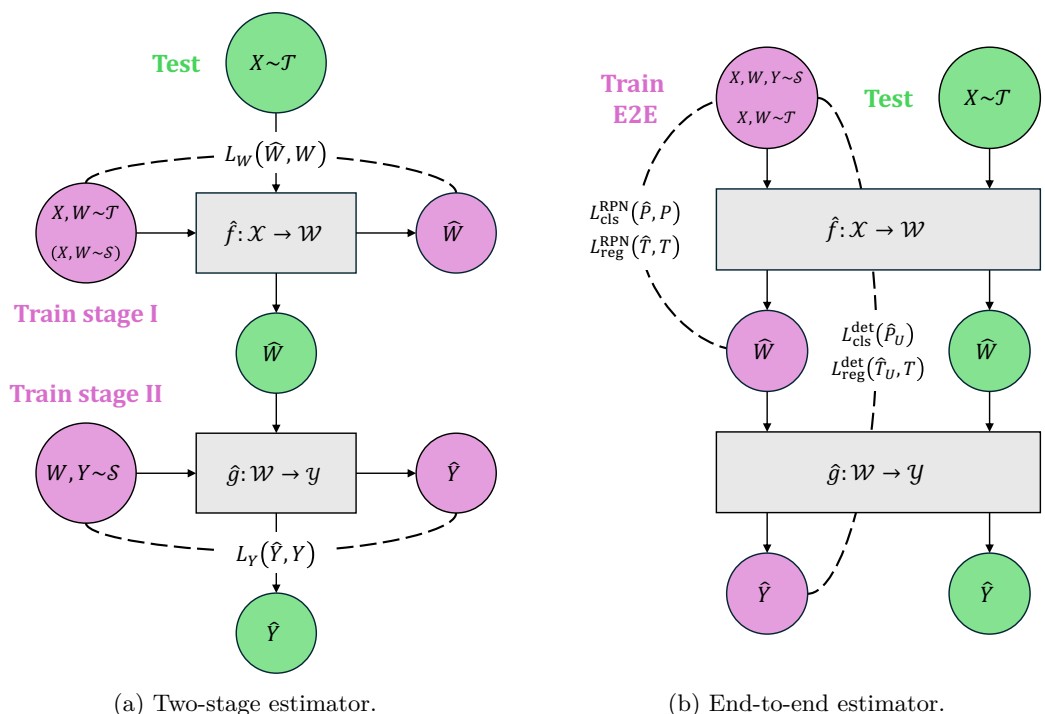

(a) Two-stage estimator.        (b) End-to-end estimator.

Figure 3: A schematic representation of the train and test flow for DALUPI using (a) the two-stage estimator presented in Section 2.2 and (b) an end-to-end architecture based on Faster R-CNN (Ren et al., 2016). In the two-stage procedure, the networks $\hat{f}$ and $\hat{g}$ are learned through empirical risk minimization of $L_W$ and $L_Y$, respectively. At test time, $\hat{f}$ and $\hat{g}$ are combined into $\hat{h} = \hat{g}(\hat{f}(X))$. The end-to-end estimator uses a region proposal network (RPN) to produce regions of interest in the input image $X$. The RPN, which serves as the network $\hat{f}$, is followed by a detection network $\hat{g}$ that predicts the class of any object within a region proposal. Training is guided by regression losses $L_{\text{reg}}^{\text{PRN}}(\hat{T}, T)$ and $L_{\text{reg}}^{\text{det}}(\hat{T}_U, T)$, as well as by classification losses $L_{\text{cls}}^{\text{PRN}}(\hat{P}, P)$ and $L_{\text{cls}}^{\text{det}}(\hat{P}_U)$. Here, $T$ and $\hat{T}$ denote ground-truth and predicted bounding box coordinates, respectively, and $\hat{T}_U$ are the predicted coordinates for a region proposal with ground-truth label $U$. Further, $\hat{P}$ is the RPN's predicted probability that a region proposal contains an object, $P$ is a binary label assigned to the proposal based on its overlap with ground-truth bounding boxes, and $\hat{P}_u$ is the probability of the ground-truth class $U$ within the proposal, as predicted by the detection network.

of one or more diseases). The partial label $y_i(k) = 1$ indicates the presence of features or objects in the image from category $k$. In our entity classification experiment, an object $j$ of class $k \in [K]$ in the image, say "Bird", will be annotated by a bounding box $t_{ij} \in \mathbb{R}^4$ surrounding the pixels of the bird, and an object label $u_{ij} = k$. In X-ray classification, $t_{ij}$ can indicate an abnormality $j$ in the X-ray image, and $u_{ij} \in \{1, \dots, K\}$ the label of the finding (e.g., "Pneumonia").

To make full use of privileged information, we train a deep neural network $\hat{h}(x) = \hat{g}(\hat{f}(x))$, where $\hat{f}$ produces *a set* of bounding box coordinates $\hat{t}_{ij}$ and extracts the pixels $\hat{w}_{ij}$ associated with each $\hat{t}_{ij}$, and where $\hat{g}$ predicts a label $\hat{u}_{ij}$ for each $\hat{w}_{ij}$. To this end, we adapt the Faster R-CNN architecture (Ren et al., 2016) which uses a region proposal network (RPN) to generate regions that are fed to a detection network for classification and refined bounding box regression. A CNN backbone in combination with the RPN region of interest pooling serves as the subnetwork $\hat{f}$, producing estimates $\hat{w}_i$ of the privileged information for an image $x_i$. For the detection network, which corresponds to the subnetwork $\hat{g}$, we use Fast-RCNN (Girshick, 2015).

Privileged information adds supervision through regression losses $L_{\text{reg}}^{\text{RPN}}(\hat{t}, t)$ and $L_{\text{reg}}^{\text{det}}(\hat{t}_u, t)$ for region proposals $\hat{t}$ and class-specific bounding box coordinates $\hat{t}_u$. We use the smooth L1 loss defined by Girshick (2015) for

both $L_{\text{reg}}^{\text{RPN}}$ and $L_{\text{reg}}^{\text{det}}$. Training is further guided by classification losses $L_{\text{cls}}^{\text{RPN}}(\hat{p}, p) = -(p \log \hat{p} + (1 - p) \log \hat{p})$ and $L_{\text{cls}}^{\text{det}}(\hat{p}_u) = -\log \hat{p}_u$, where $\hat{p}$ is the RPN's predicted probability that a region proposal contains an object, $p$ is a binary label assigned to the proposal based on its overlap with ground-truth bounding boxes, and $\hat{p}_u$ is the probability of the ground-truth class $u$ within the proposal, as predicted by the detection network.

In Appendix A.2, we provide details of the learning objective and architecture and describe small modifications to the training procedure of Faster R-CNN to accommodate the DALUPI setting. Unlike the two-stage estimator, we train Faster R-CNN (both $\hat{f}$ and $\hat{g}$) end-to-end, minimizing both losses at once. In entity classification experiments (see Table 3 and Figure 5), we also train this model in a LUPI setting, where *no* information from the target domain is used, but privileged information from the source domain is used.

## 4 Experiments

We evaluate the empirical benefits of learning using privileged information, compared to the other data availability settings in Table 1, across four UDA image classification tasks where PI is available in the forms described in Section 3. Widely used datasets for UDA evaluation like OfficeHome (Venkateswara et al., 2017) and large-scale benchmark suites like DomainBed (Gulrajani & Lopez-Paz, 2021), VisDA (Peng et al., 2017) and WILDS (Koh et al., 2021) *do not* include privileged information and cannot be used for evaluation here. Thus, we first compare our method to baselines on the recent CelebA task (Xie et al., 2020) which includes PI in the form of binary attributes (Section 4.1). Additionally, we propose three new tasks based on well-known image classification data sets with regions of interest as PI (Section 4.2–4.4). In Section 4.1 and 4.2, we use the two-stage estimator with the subnetwork $\hat{f}$ based on the ResNet-18 architecture (He et al., 2016a). In Section 4.3 and 4.4, we use our variant of Faster R-CNN with a ResNet-50 backbone.

Our goal is to collect evidence that DALUPI improves adaptation bias and sample efficiency compared to methods that do not make use of PI. We choose baselines to illustrate these two disparate settings. First, we compare DALUPI to supervised learning baselines, SL-S and SL-T, trained on labeled examples from the source and target domain, respectively. SL-S is a simple but strong baseline: On benchmark suites like DomainBed and WILDS, there is still no UDA method that *consistently* outperforms SL-S (ERM) without transfer learning (Gulrajani & Lopez-Paz, 2021; Koh et al., 2021). SL-T serves as an oracle comparison since it uses labels from the target domain which are normally unavailable in UDA. Second, we compare DALUPI to two UDA methods—domain adversarial neural networks (DANN) (Ganin et al., 2016) and the margin disparity discrepancy (MDD) (Zhang et al., 2019)—which have theoretical guarantees but do not make use of PI. These baselines are all based on the ResNet architecture. In Section 4.1, we compare DALUPI also to In-N-Out (Xie et al., 2020), which was designed to make use of auxiliary (privileged) attributes for training domain adaptation models. We do not include this model in other experiments as it was not designed to use regions of interest as privileged information. The exact architectures of all models and baselines are described in Appendix A, along with details on experimental setup and hyperparameters.

For each task and task-specific setting (label skew, amount of privileged information, etc.), we train 10 models from each relevant class using hyperparameters randomly selected from given ranges (see Appendix A). For DANN and MDD, the trade-off parameter, which regularizes domain discrepancy in representation space, increases from 0 to 0.1 during training; for MDD, the margin parameter is set to 3. All models are evaluated on a held-out validation set from the source domain and the best-performing model in each class is then evaluated on held-out test sets from both domains. For SL-T, we use a held-out validation set from the target domain. We repeat this procedure over 5 or 10 seeds, controlling the data splits and the random number generation. We report accuracy and area under the ROC curve (AUC) with 95 % confidence intervals computed by bootstrapping over the seeds.

### 4.1 Celebrity Photo Classification With Binary Attributes as PI

In the case where privileged information is available as binary attributes, we follow Xie et al. (2020) who introduced a binary classification task based on the CelebA dataset (Liu et al., 2015), where the goal is to predict whether the person in an image has been identified as male or female ($Y$) in one of the binary

Table 2: Celebrity photo classification. DALUPI performs comparably to the In-N-Out models in Xie et al. (2020). Note: In-N-Out results are reported as the average of 5 trials with 90 % confidence intervals.

|  | Target accuracy |
|---|---|
| SL-T | 86.6 (86.3, 86.9) |
| SL-S | 78.4 (77.1, 80.0) |
| DANN | 78.2 (76.2, 80.3) |
| MDD | 78.3 (77.5, 79.1) |
| In-N-Out (w/o pretraining) | 78.5 (77.2, 79.9) |
| In-N-Out (w. pretraining) | 79.4 (78.7, 80.1) |
| In-N-Out (rep. self-training) | 80.4 (79.7, 81.1) |
| DALUPI ($W \sim \mathcal{T}$) | 76.4 (73.8, 78.6) |
| DALUPI ($W \sim \mathcal{S}, \mathcal{T}$) | 80.3 (77.9, 82.7) |

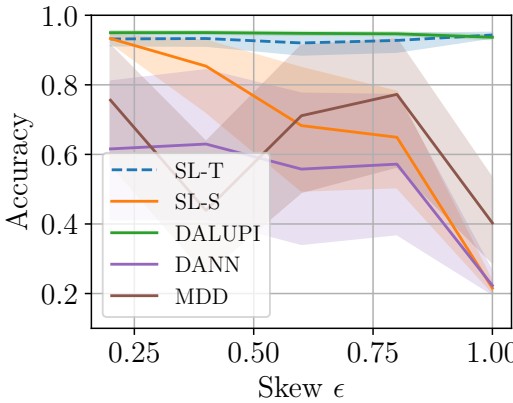

Figure 4: Digit classification. Target domain accuracy as a function of association $\epsilon$ between background and label in $\mathcal{S}$. As the skew increases, the target-domain performance of the non-privileged models deteriorates.

attributes that accompanies the data set's photos of celebrities ($X$). Like Xie et al. (2020), we use 7 of the 40 other attributes (`Bald`, `Bangs`, `Mustache`, `Smiling`, `5_o_Clock Shadow`, `Oval_Face`, and `Heavy_Makeup`) as a vector of privileged information $W \in \{0, 1\}^7$. The target and source domains are defined by people wearing ($\mathcal{T}$) and not wearing ($\mathcal{S}$) a hat. The respective datasets contain 3,000 and 2,000 images. An extra 30,000 unlabeled source samples are available to train estimators (DALUPI and In-N-Out) that can utilize privileged information from both source and target. More details can be found in (Xie et al., 2020) and in Appendix A.3.

Table 2 shows the target accuracy for each model. We observe that when DALUPI is provided with PI from both source and target, it performs comparably to the best-performing In-N-Out model proposed by Xie et al. (2020), while outperforming other feasible baselines on average. Confidence intervals overlap for all feasible models. Notably, the best-performing In-N-Out models require four or more rounds of training to achieve their results (baseline, auxiliary input, auxiliary output pre-training, tuning and self-training) (Xie et al., 2020). Both DALUPI and In-N-Out benefit from access to privileged information from both the source and target domain (pre/self-training for In-N-Out).

Finally, it is worth noting that neither covariate shift, nor sufficiency are likely to hold with respect to $W$ in this task. Specifically, photos with none of the 7 attributes active, $w = \mathbf{0}$, have different label rates and majority label in $\mathcal{S}$ and $\mathcal{T}$ (the rates of labels are $\bar{Y}_{\mathcal{S}} = 0.64$ and $\bar{Y}_{\mathcal{T}} = 0.46$, respectively) and therefore $P(Y|W)$ is not constant, i.e. covariate shift is violated. In addition, the best model we have found trained on $W$ alone achieves only 65 % accuracy, compared to the results in Table 2—sufficiency is unlikely to hold. Thus, DALUPI is robust to violations of these assumptions.

## 4.2 Digit Classification With Single Region of Interest as PI

We construct a synthetic image dataset, based on the assumptions of Proposition 1, to verify that there are problems where DALUPI is guaranteed successful transfer but standard UDA is not. Starting from CIFAR-10 (Krizhevsky, 2009) images upscaled to $128 \times 128$, we insert a random $28 \times 28$ digit image from the MNIST dataset (Lecun, 1998), with a label in the range 0–4, into a random location of each CIFAR-10 image, forming the input image $X$ (see Figure 1 (top) for examples). The label $Y \in \{0, \ldots, 4\}$ is determined by the MNIST digit. We store the bounding box around the inserted digit image and use the pixels contained within it as privileged information $W$ during training. The domains are constructed using CIFAR-10's first five and last five classes as source and target backgrounds, respectively. Both source and target datasets contain 15,298 images each. To increase the difficulty of the task, we make the digit be the mean color of the

Table 3: Entity classification. UDA models have access to all unlabeled target samples, LUPI to all PI (source), and DALUPI to all PI (source and target).

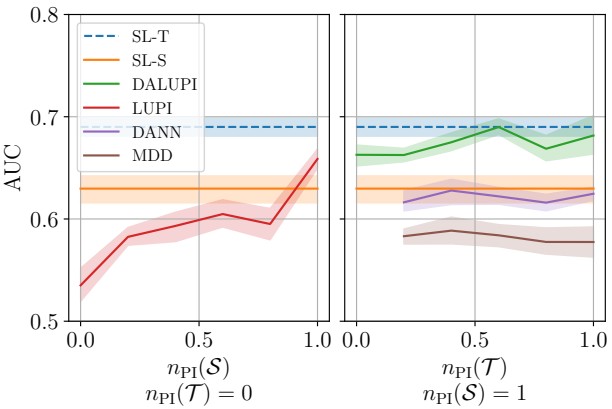

Figure 5: Entity classification. Target domain AUC. The performance of SL-S and SL-T is extended across the x-axes for visual purposes. DANN and MDD use an increasing fraction of target samples $\tilde{x}$ but no PI.

|        | Source AUC        | Target AUC        |
|--------|-------------------|-------------------|
| SL-T   | 60.1 (58.7, 61.5) | 69.0 (68.1, 69.9) |
| SL-S   | 69.5 (68.6, 70.4) | 63.0 (61.6, 64.2) |
| DANN   | 68.1 (67.5, 68.7) | 62.5 (61.9, 63.1) |
| MDD    | 62.4 (61.1, 63.9) | 57.7 (56.3, 59.2) |
| LUPI   | 69.3 (68.5, 70.1) | 65.9 (65.0, 66.8) |
| DALUPI | 71.4 (70.3, 72.4) | 68.2 (66.3, 70.1) |

dataset and make the digit background transparent so that the border of the image is less distinct. This may slightly violate Assumption 2 with respect to the region of interest $W$ since the backgrounds differ between domains.

To understand how successful transfer depends on domain overlap and access to sufficient privileged information, we include a *skew parameter* $\epsilon \in [\frac{1}{c}, 1]$, where $c = 5$ is the number of digit classes, which determines the correlation between digits and backgrounds. For a source image $i$ with digit label $Y_i \in \{0, \ldots, 4\}$, we select a random CIFAR-10 image with class $B_i \in \{0, \ldots, 4\}$ with probability $P(B_i = b \mid Y_i = y) = \{\epsilon, \text{ if } b = y; \ (1 - \epsilon)/(c - 1), \text{otherwise}\}$. For target images, digits and backgrounds are matched uniformly at random. The choice $\epsilon = \frac{1}{c}$ yields a uniform distribution and $\epsilon = 1$ is equivalent to the background carrying as much signal as the privileged information. We hypothesize that $\epsilon = 1$ is the worst possible case where confusion of the model is likely, which would lead to poor adaptation under domain shift.

In Figure 4, we observe the conjectured behavior. As the skew $\epsilon$ and the association between background and label increases, the performance of SL-S decreases rapidly on the target domain. At $\epsilon = 1$, it performs no better than random guessing, likely because the model has learned to associate spurious features in the background with the label of the digit. We also observe that DANN and MDD deteriorate in performance with increased correlation between the label and the background. In contrast, DALUPI is unaffected by the skew as the subset of pixels extracted by $\hat{f}$ only carries some of the background with it, while containing sufficient information to make good predictions. Interestingly, DALUPI also seems to be as good or slightly better than the oracle SL-T in this setting. This may be due to improved sample efficiency from using PI.

### 4.3 Entity Classification With Multiple Regions of Interest as PI

Next, we consider multi-label classification of the presence of four types of entities (persons, cats, dogs, and birds) indicated by a binary vector $Y \in \{0, 1\}^4$ for images $X$ from the MS-COCO dataset (Lin et al., 2014). PI is used to localize regions of interest $W$ related to the entities, provided as bounding box annotations. We define source and target domains $\mathcal{S}$ and $\mathcal{T}$ as indoor and outdoor images, respectively. Indoor images are extracted by filtering out images from the MS-COCO super categories "indoor" and "appliance" that also contain at least one of the four main label classes. Outdoor images are extracted using the super categories "vehicle" and "outdoor". In total, there are 5,231 images in the source and 5,719 images in the target domain; the distribution of labels is provided in Appendix A.5.

Sufficiency is likely to hold in this task because the pixels contained in a bounding box should be sufficient for an annotator to classify the entity according to the four categories above, irrespective of the pixels outside

Table 4: X-ray task. Test AUC for the three pathologies in the target domain for all considered models. Boldface indicates the best-performing feasible model; SL-T uses target labels.

| | ATL | CM | PE |
|---|---|---|---|
| SL-T | 57 (56, 58) | 59 (55, 63) | 79 (78, 80) |
| SL-S | **55 (55, 56)** | 61 (58, 64) | 73 (70, 75) |
| DANN | 53 (51, 55) | 55 (53, 58) | 55 (51, 61) |
| MDD | 49 (48, 50) | 51 (51, 52) | 51 (48, 54) |
| DALUPI | **55 (55, 56)** | **72 (71, 73)** | **74 (72, 76)** |

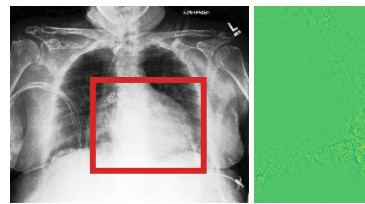

Figure 6: Left: Example from the X-ray target test set with label CM. The red rectangle indicates the bounding box predicted by DALUPI. Right: saliency map for CM for SL-S.

of the box. Similarly, covariate shift is likely to hold since the label attributed to the pixels in a bounding box should be the same, whether the entity is indoor or outdoor.

We study the effect of adding privileged information by first training the end-to-end model in a LUPI setting, using all $(x, y)$ samples from the source domain and increasing the fraction of inputs for which PI is available, $n_{\mathrm{PI}}(\mathcal{S})$, from 0 to 1. We then train the model in a DALUPI setting, increasing the fraction of $(\tilde{x}, \tilde{w})$ samples from the target domain, $n_{\mathrm{PI}}(\mathcal{T})$, from 0 to 1, while keeping $n_{\mathrm{PI}}(\mathcal{S}) = 1$. We train SL-S and SL-T using all available data and increase the fraction of unlabeled target samples used by DANN and MDD from 0.2 to 1 while using all data from the source domain.

Table 3 shows the models' source and target domain AUC, averaged over the four entity classes, when the UDA models have access to all unlabeled target samples, LUPI to all PI from the source domain, and DALUPI to all PI from both domains. Clearly, DALUPI yields a substantial gain in adaptation. As we see in Figure 5, the performance of LUPI increases as $n_{\mathrm{PI}}(\mathcal{S})$ increases. When additional $(\tilde{x}, \tilde{w})$ samples from the target domain are added, DALUPI outperforms SL-S and approaches the performance of SL-T. We note that DANN and MDD do not benefit as much from added unlabeled target samples as DALUPI does. Their weak performance could be explained by difficulties in adversarial training. The gap between LUPI and SL-S for $n_{\mathrm{PI}}(\mathcal{S}) = 0$ is anticipated; we do not expect the detection network to work well without bounding box supervision.

## 4.4 X-ray Classification With Multiple Regions of Interest as PI

As a real-world application, we study detection of pathologies in chest X-ray images. We use the ChestX-ray8 dataset (Wang et al., 2017) as source domain and the CheXpert dataset (Irvin et al., 2019) as target domain.[1] As PI, we use the regions of pixels associated with each found pathology, as annotated by domain experts using bounding boxes. For the CheXpert dataset, only pixel-level segmentations are available, and we create bounding boxes that tightly enclose the segmentations. It is not obvious that the pixels within such a bounding box are sufficient for classifying the pathology. For this reason, we suspect that some of the assumptions of Proposition 1 may be violated. However, as we find below, DALUPI improves empirical performance compared to baselines for small training sets, thereby demonstrating increased sample efficiency.

We consider the three pathologies that exist in both datasets and for which there are annotated findings: atelectasis (ATL: collapsed lung), cardiomegaly (CM: enlarged heart), and pleural effusion (PE: water around the lung). There are 457 and 118 annotated images in the source and target domain, respectively. We train DALUPI, DANN and MDD using all these images. SL-S is trained with the 457 source images and SL-T with the 118 target images as well as 339 labeled but non-annotated target images. Neither SL-S, SL-T, DANN, nor MDD support using privileged information. The distributions of labels and bounding box annotations are given in Appendix A.6.

---

[1]This study was granted IRB approval.

In Table 4, we present the per-class AUCs in the target domain. DALUPI outperforms all baseline models, including the target oracle, in detecting CM. For ATL and PE, it performs similarly to or better than the other feasible models. That SL-T is better at predicting PE is not surprising because this pathology is most prevalent in the target domain. In Figure 6, we show a single-finding image from the target test set with ground-truth label CM. The predicted bounding box of DALUPI with the highest probability is added to the image. DALUPI identifies the region of interest (the heart) and makes a correct classification. The rightmost panel shows the saliency map for the ground truth class for SL-S. We see that the gradients are mostly constant, indicating that the model is uncertain. In Appendix B, we show AUC for CM for the models trained with additional examples *without* bounding box annotations. We find that SL-S reaches the performance of DALUPI when a large amount of labeled examples are provided. This indicates that identifiability is not the main obstacle for adaptation and that PI improves sample efficiency.

## 5 Related Work

Learning using privileged information was first introduced by Vapnik & Vashist (2009) for support vector machines (SVMs), and was later extended to empirical risk minimization (Pechyony & Vapnik, 2010). Methods using PI, which is sometimes called hidden information or side information, has since been applied in many diverse settings such as healthcare (Shaikh et al., 2020), finance (Silva et al., 2010), clustering (Feyereisl & Aickelin, 2012) and image recognition (Vu et al., 2019; Hoffman et al., 2016). Related concepts include knowledge distillation (Hinton et al., 2015; Lopez-Paz et al., 2016), where a teacher model trained on additional variables adds supervision to a student model, and weak supervision (Robinson et al., 2020) where so-called weak labels are used to learn embeddings, subsequently used for the task of interest. Furthermore, in the realm of NLP, there is the related concept of learning using feature feedback, where additional annotations that are related to the associated task label are provided (Katakkar et al., 2022; Kaushik et al., 2021). These works are mostly of an empirical nature, and theoretical work on the subject either considers linear models/SVMs (Poulis & Dasgupta, 2017) or a teacher/student-type setup where additional supervision is given when the model predicts incorrectly (Dasgupta et al., 2018). The use of PI for deep image classification has been investigated by Chen et al. (2017) and Han et al. (2023) but these works only cover regular supervised learning where source and target domains coincide. Further, Sharmanska et al. (2014) used regions of interest in images as privileged information to improve the accuracy of image classifiers, but did not consider domain shift either.

Domain adaptation using PI has been considered before with SVMs (Li et al., 2022; Sarafianos et al., 2017), but not with more complex classifiers such as neural networks. Vu et al. (2019) used scene depth as PI in semantic segmentation using deep neural networks. However, they only used PI from the source domain and they did not provide any theoretical analysis. Xie et al. (2020) provide some theoretical results for a similar setup to ours. However, these are specifically for linear classifiers while our approach holds for any type of classifier. Motiian (2019) investigated PI and domain adaptation using the information bottleneck method for visual recognition. However, their setting differs from ours in that each observation comprises source-domain and target-domain features, a label and PI. Another related approach is that of subsidiary tasks (Kundu et al., 2022; Ye et al., 2022). However, in these settings the additional tasks performed are used to build a representation that helps with the main task through domain alignment. Our approach instead seeks to use information which directly relates to the main task.

## 6 Discussion

We have presented DALUPI: unsupervised domain adaptation by learning using privileged information (PI). The framework provides provable guarantees for adaptation under relaxed assumptions on the input features, at the cost of collecting a larger variable set, such as attribute or bounding box annotations, during training. Our analysis inspired practical algorithms for image classification which we evaluated using three kinds of privileged information. In our experiments, we demonstrated tasks where our approach is successful while existing adaptation methods fail. We observed empirically also that methods using privileged information are more sample-efficient than comparable non-privileged learners, in line with the literature. In fact, DALUPI

models occasionally even outperform oracle models trained using target labels due to their sample efficiency. Thus, we recommend considering these methods in small-sample settings.

The main contribution of the paper is the proposed learning paradigm for domain adaptation with privileged information. Since common benchmark datasets in UDA lack privileged information related to the learning problem, we created three new tasks for evaluating our framework, see Section 4.2–4.4, which itself is a notable contribution. We hope that this work inspires the community to develop additional datasets for UDA using privileged information.

To avoid assuming that domain overlap is satisfied with respect to input covariates, we require that the label is conditionally independent of the input features given the PI—that the PI is "sufficient". This is a limitation whenever sufficiency is difficult to verify. However, in our motivating example of image classification, a domain expert could *choose* PI so that sufficiency is reasonably justified. Moreover, in experiments on CelebA, we see empirical gains from our approach even when sufficiency is known to be violated. Another limitation is that we still rely on overlap in the privileged information, $W$, which may also be violated in some circumstances. It is more likely that overlap holds for $W$ when, for example, it is a subset of $X$, as argued in Figure 2. Designing experiments to test how sensitive DALUPI is to violations of these assumptions is an interesting direction for future work.

The use of regions of interest as privileged information brings up an interesting point concerning the relationship between the label and the privileged information. In object detection tasks, it is natural to treat the bounding box coordinates as label information. In this work, however, the learning tasks were multi-class and multi-label image classification, not object detection. Producing a perfect box $W$ was not the goal of the learning task, and the bounding boxes were therefore neither critical for the task nor for the labels. Instead, the bounding boxes were privileged information and our experiments in Section 4.2–4.4 sought to quantify the value of this added information, compared to not having it. Therefore, we compared our method to image classification baselines. It is not obvious a priori that learning from object locations improves the adaptation of image classifiers.

If there is a lack of PI available to the models one might mitigate this by either 1) using the limited amount of PI that is available to learn $\hat{g}$ and assume that it is good enough to achieve reasonable overall performance; or 2) using the learned $\hat{f}$ to create "weak" PI labels for the inputs that are missing PI, similar to the work of e.g. Robinson et al. (2020). However, one should note that the latter approach might bias the model in unintended ways and, as such, should be undertaken with some caution.

In future work, our framework could be applied to a more diverse set of tasks, with different modalities of inputs and privileged information to investigate if the findings here can be replicated and extended. Moreover, such work could consider different types and degrees of shifts to further corroborate the stability and resistance to noise which we observe here. More broadly, using PI may be viewed as "building in" domain knowledge in the structure of the adaptation problem and we see this as a promising direction for further research.

### Acknowledgments

This work was partially supported by the Wallenberg AI, Autonomous Systems and Software Program (WASP) funded by the Knut and Alice Wallenberg Foundation

The computations and data handling were enabled by resources provided by the National Academic Infrastructure for Supercomputing in Sweden (NAISS), partially funded by the Swedish Research Council through grant agreement no. 2022-06725.

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

## A Experimental Details

In this section, we give further details of the experiments. All code is written in Python and we mainly use PyTorch in combination with skorch (Tietz et al., 2017) for our implementations of the net-works. For Faster R-CNN, we adapt the implementation provided by torchvision through the func-tion `fasterrcnn_resnet50_fpn`. For DANN and MDD, we use the ADAPT TensorFlow implementation (de Mathelin et al., 2021) with a ResNet-50-based encoder. We initially set the trade-off parameter $\lambda$, which controls the amount of domain adaption regularization, to 0 and then increase it to 0.1 in 10,000 gradient steps according to the formula $\lambda = \beta(2/(1 + e^{-p}) - 1)/C$, where $p$ increases linearly from 0 to 1, $\beta$ is a parameter specified for each experiment, and $C = 2/(1 + e^{-1}) - 1$. For MDD, we fix the margin parameter $\gamma$ to 3. The source and target baselines are based on the ResNet-50 architecture when PI is provided as multiple regions of interest; otherwise, the ResNet-18 architecture is used. The architecture of DALUPI in each experiment is specified in the respective subsection below.

We use the Adam optimizer in all experiments. Learning rate decay is treated as a hyperparameter. For ADAPT models (DANN and MDD), the learning rate is either constant or decayed according to $\mu_0/(1 + \alpha p)^{3/4}$, where $\mu_0$ is the initial learning rate, $p$ increases linearly from 0 to 1, and $\alpha$ is a parameter specified in each experiment (see below). For non-ADAPT models, the learning rate is either constant or decayed by a factor 0.1 every $n$th epoch, where $n$ is another hyperparameter.

For all models except LUPI and DALUPI, the classifier network following the encoder is a simple MLP with two possible settings: Either it is a single linear layer from inputs to outputs or a three-layer network with ReLU activations between the layers. This choice is treated as a hyperparameter in our experiments. The nonlinear case has the following structure where $n$ is the number of input features:

- fully connected layer with $n$ neurons

- ReLU activation layer

- fully connected layer with $n//2$ neurons

- ReLU activation layer

- fully connected layer with $n//4$ neurons.

All models were trained using NVIDIA Tesla A40 GPUs and the development and evaluation of this study required approximately 30,000 hours of GPU training. The code is available on GitHub: `https://github.com/Healthy-AI/dalupi`.

### A.1 DALUPI With Two-stage Classifier

Here, we describe in more detail how we construct our two-stage classifier for image classification when privileged information is provided as a single region of interest as in the digit classification task (Section 4.2). When privileged information is provided as binary attributes, we can directly learn the two-stage estimator according to Equation 2. In this task, it was found that using the cross entropy loss and using continuous outputs from $f$ provided superior performance compared to other losses. In the digit classification task, each image $x_i$ has a single label $y_i \in \{0, \dots, 4\}$ determined by the MNIST digit. Privileged information is given by a single bounding box with coordinates $t_i \in \mathbb{R}^4$ enclosing a subset of pixels $w_i$ corresponding to the digit. The training procedure is summarized in Algorithm 1 and further described below.

We first learn $\hat{d}$ which is a function that takes target image data, $\tilde{x}_i$, and bounding box coordinates, $t_i$, and learns to output bounding box coordinates, $\hat{t}_i$, which should contain the privileged information $w_i$. Note that we do not exactly follow the setup in Equation 2 since we do not need to actually predict the pixel values within the bounding box. If we find a good enough estimator of $t_i$ we should minimize the loss of $f$ in Equation 2. To obtain the privileged information we apply a deterministic function $\phi$ which crops and scales an image using the associated bounding box, $t_i$. We can now write the composition of these two functions as $\hat{f}(x_i) = \phi(x_i, \hat{d}(x_i))$ which outputs the privileged information. The function $\phi$ is hard-coded and therefore not learned.

In the second step, we learn $\hat{g}$ to predict the label from the privileged information $w_i$, which is a cropped version of $x_i$ where the cropping is defined by the bounding box $t_i$ around the digit. This cropping and resizing is performed by $\phi$. When we evaluate the performance of this classifier we combine the two models into one, $\hat{h}(x) = \hat{g}(\phi(x, \hat{d}(x)))$. We use the mean squared error loss for learning $\hat{d}$ and categorical cross-entropy (CCE) loss for $\hat{g}$.

---

**Algorithm 1** Training of the two-stage model.

---

1: **procedure** Two_stage $(\tilde{x}_i, w_i, t_i, y_i)$
2:     Empirically minimize $\frac{1}{m} \sum_{i=1}^{m} \|d(\tilde{x}_i) - t_i\|^2$ and obtain $\hat{d}$.
3:     Empirically minimize $\frac{1}{n} \sum_{i=1}^{n} CCE(g(w_i), y_i)$ and obtain $\hat{g}$.
4:     Compose $\hat{d}$, $\hat{g}$ and $\phi$ into $\hat{h}(x) = \hat{g}(\phi(x, \hat{d}(x)))$.
5: **end procedure**

---

### A.2 DALUPI With Faster R-CNN

For multi-label classification, we adapt Faster R-CNN (Ren et al., 2016) outlined in Figure 7 and described below. Faster R-CNN uses a region proposal network (RPN) to generate region proposals which are fed to a detection network for classification and bounding box regression. This way of solving the task in subsequent steps has similarities with our two-stage algorithm although Faster R-CNN can be trained end-to-end. We make small modifications to the training procedure of the original model in the end of this section.

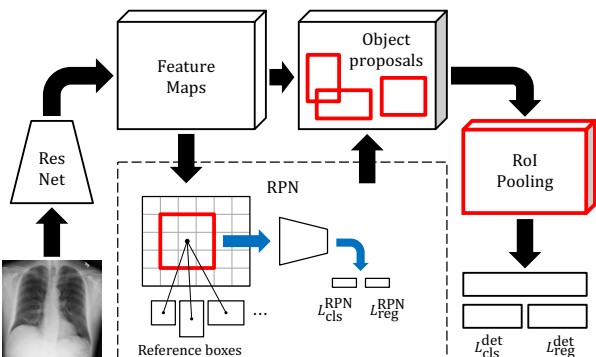

Figure 7: Faster R-CNN (Ren et al., 2016) architecture. The RoI pooling layer and the classification and regression layers are part of the Fast R-CNN detection network (Girshick, 2015).

The RPN generates region proposals relative to a fixed number of reference boxes—anchors—centered at the locations of a sliding window moving over convolutional feature maps. Each anchor is assigned a binary label $p \in \{0, 1\}$ based on its overlap with ground-truth bounding boxes; positive anchors are also associated with a ground-truth box with location $t$. The RPN loss for a single anchor is

$$L^{\mathrm{RPN}}(\hat{p}, p, \hat{t}, t) := L_{\mathrm{cls}}^{\mathrm{RPN}}(\hat{p}, p) + p L_{\mathrm{reg}}^{\mathrm{RPN}}(\hat{t}, t), \tag{3}$$

where $\hat{t}$ represents the refined location of the anchor and $\hat{p}$ is the estimated probability that the anchor contains an object. The binary cross-entropy loss and a smooth $L_1$ loss are used for the classification loss $L_{\mathrm{cls}}^{\mathrm{RPN}}$ and the regression loss $L_{\mathrm{reg}}^{\mathrm{RPN}}$, respectively. For a mini-batch of images, the total RPN loss is computed based on a subset of all anchors, sampled to have a ratio of up to 1:1 between positive and negative ditto.

A filtered set of region proposals are projected onto the convolutional feature maps. For each proposal, the detection network—Fast R-CNN (Girshick, 2015)—outputs a probability $\hat{p}(k)$ and a predicted bounding box location $\hat{t}(k)$ for each class $k$. Let $\hat{p} = (\hat{p}(0), \ldots, \hat{p}(K))$, where $\sum_k \hat{p}(k) = 1$, $K$ is the number of classes and 0 represents a catch-all background class. For a single proposal with ground-truth coordinates $t$ and multi-class label $u \in \{0, \ldots, K\}$, the detection loss is

$$L^{\mathrm{det}}(\hat{p}, u, \hat{t}_u, t) = L_{\mathrm{cls}}^{\mathrm{det}}(\hat{p}, u) + \mathbf{I}_{u \geq 1} L_{\mathrm{reg}}^{\mathrm{det}}(\hat{t}_u, t), \tag{4}$$

where $L_{\mathrm{cls}}^{\mathrm{det}}(\hat{p}, u) = -\log \hat{p}(u)$ and $L_{\mathrm{reg}}^{\mathrm{det}}$ is a smooth $L_1$ loss. To obtain a probability vector for the entire image, we maximize, for each class $k$, over the probabilities of all proposals.

During training, Faster R-CNN requires that all input images $x$ come with at least one ground-truth annotation (bounding box) $w$ and its corresponding label $u$. To increase sample-efficiency, we enable training the model using non-annotated but labeled samples $(x, y)$ from the source domain and annotated but unlabeled samples $(\tilde{x}, \tilde{w})$ from the target domain. In the RPN, no labels are needed, and we simply ignore anchors from non-annotated images when sampling anchors for the loss computation. For the computation of Equation 4, we handle the two cases separately. We assign the label $u = -1$ to all ground-truth annotations from the target domain and multiply $L_{\mathrm{cls}}^{\mathrm{det}}$ by the indicator $\mathbf{I}_{u \geq 0}$. For non-annotated samples $(x, y)$ from the source domain, there are no box-specific coordinates $t$ or labels $u$ but only the labels $y$ for the entire image. In this case, 4 is undefined and we instead compute the binary cross-entropy loss between the per-image label and the probability vector for the entire image.

We train the RPN and the detection network jointly as described in Ren et al. (2016). To extract feature maps, we use a Feature Pyramid Network (Lin et al., 2017) on top of a ResNet-50 architecture He et al. (2016b). We use the modified model in the experiments in Section 4.3 and 4.4. In Section 4.3, we also train this model in a LUPI setting, where no information from the target domain is used.

### A.3 Celebrity Photo Classification With Binary Attribute Vector

In our experiment based on CelebA (Liu et al., 2015), the input $x$ is an RGB image which has been resized to 64×64 pixels, the target $y$ is a binary label for gender of the subject of the image, and the privileged information $w$ are 7 binary-valued attributes. The attributes used in this experiment are: `Bald`, `Bangs`, `Mustache`, `Smiling`, `5_o_Clock_Shadow`, `Oval_Face` and `Heavy_Makeup`. We use a subset of the CelebA dataset with 2,000 labeled source examples and 3,000 unlabeled target examples. We use 1,000 samples each for the source validation set, source test set, and target test set, respectively. The target oracle, SL-T, is trained using labels provided for the 3,000 target examples, with 20 % of these examples set aside for validation. The same unlabeled validation set is used to validate the first DALUPI network, $\hat{f}$. When using privileged information from the source domain to train $\hat{f}$, we use 30,000 extra samples $(x, w)$ with PI.

For DALUPI, we use the two-stage estimator with the network $\hat{f}$ based on ResNet-18 followed by a non-linear MLP. The network $\hat{g}$ is an MLP with two hidden layers of with 256 neurons each. We train the models for 100 epochs. If the validation accuracy (or validation AUC for $\hat{f}$) does not improve for 10 subsequent epochs, we stop the training earlier. For DALUPI, the early stopping patience is 15 for each network. We treat the problem as multi-class classification with two classes and use the categorical cross entropy loss for SL-S, SL-T, DANN, and MDD.

#### A.3.1 Hyperparameters

We randomly choose hyperparameters from the following predefined sets of values:

- SL-S and SL-T:
    - batch size: $(16, 32, 64)$
    - initial learning rate: $(1.0 \times 10^{-4}, 1.0 \times 10^{-3})$
    - step size $n$ for learning rate decay: $(15, 30, 100)$
    - weight decay: $(1.0 \times 10^{-4}, 1.0 \times 10^{-3})$
    - dropout (encoder): $(0, 0.1, 0.2, 0.5)$
    - nonlinear classifier: (`True`, `False`).

- DALUPI:
    - batch size: $(16, 32, 64)$
    - initial learning rate: $(1.0 \times 10^{-5}, 1.0 \times 10^{-4}, 1.0 \times 10^{-3})$
    - step size $n$ for learning rate decay: $(15, 30, 100)$
    - weight decay: $(1.0 \times 10^{-4}, 1.0 \times 10^{-3})$.

- DANN:
    - batch size: $(16, 32, 64)$
    - initial learning rate: $(1.0 \times 10^{-4}, 1.0 \times 10^{-3})$
    - parameter $\alpha$ for learning rate decay: $(0, 1.0)$
    - weight decay: $(1.0 \times 10^{-4}, 1.0 \times 10^{-3})$
    - dropout (encoder): $(0, 0.1, 0.2, 0.5)$
    - width of discriminator network: $(64, 128, 256)$
    - depth of discriminator network: $(2, 3)$
    - nonlinear classifier: (`True`, `False`)
    - parameter $\beta$ for adaption regularization decay: $(0.1, 1.0, 10.0)$.

- MDD:
    - batch size: $(16, 32, 64)$
    - initial learning rate: $(1.0 \times 10^{-4}, 1.0 \times 10^{-3})$

- parameter $\alpha$ for learning rate decay: $(0, 1.0)$
- weight decay: $(1.0 \times 10^{-4}, 1.0 \times 10^{-3})$
- dropout (encoder): $(0, 0.1, 0.2, 0.5)$
- nonlinear classifier: (`True`, `False`)
- maximum norm value for classifier weights: $(0.5, 1.0, 2.0)$
- parameter $\beta$ for adaption regularization decay: $(0.1, 1.0, 10.0)$.

## A.4 Digit Classification With Single Bounding Box as PI

In the digit classification task, we separate $20\,\%$ of the available source and target data into a test set. We likewise use $20\,\%$ of the training data for validation purposes. For DALUPI we use ResNet-18 for the function $\hat{f}$. We replace the default fully connected layer with a fully connected layer with 4 neurons to predict the coordinates of the bounding box. The predicted bounding box is resized to a $28 \times 28$ square no matter the initial size. We use a simple convolutional neural network for the function $\hat{g}$ with the following structure:

- convolutional layer with 16 output channels, kernel size of 5, stride of 1, and padding of 2
- max pooling layer with kernel size 2, followed by a ReLU activation
- convolutional layer with 32 output channels, kernel size of 5, stride of 1, and padding of 2
- max pooling layer with kernel size 2, followed by a ReLU activation
- dropout layer with $p = 0.4$
- fully connected layer with 50 out features, followed by ReLU activation
- dropout layer with $p = 0.2$
- fully connected layer with 5 out features.

The model training is stopped when the best validation accuracy (or validation loss for $\hat{f}$) does not improve over 10 epochs or when the model has been trained for 100 epochs, whichever occurs first. All models are trained from scratch, without pretrained weights. We use the categorical cross entropy loss for SL-S, SL-T, DANN, and MDD.

### A.4.1 Hyperparameters

We randomly choose hyperparameters from the following predefined sets of values:

- SL-S and SL-T:
  - batch size: $(16, 32, 64)$
  - initial learning rate: $(1.0 \times 10^{-4}, 1.0 \times 10^{-3})$
  - step size $n$ for learning rate decay: $(15, 30, 100)$
  - weight decay: $(1.0 \times 10^{-4}, 1.0 \times 10^{-3})$
  - dropout (encoder): $(0, 0.1, 0.2, 0.5)$
  - nonlinear classifier: (`True`, `False`).

- DALUPI:
  - batch size: $(16, 32, 64)$
  - initial learning rate: $(1.0 \times 10^{-4}, 1.0 \times 10^{-3})$
  - step size $n$ for learning rate decay: $(15, 30, 100)$
  - weight decay: $(1.0 \times 10^{-4}, 1.0 \times 10^{-3})$.

Table 5: Marginal label distribution in source and target domains for the entity classification task based on the MS-COCO dataset. The background class contains images where none of the four entities are present.

| Domain | Person | Dog | Cat | Bird | Background |
|--------|--------|-----|------|------|------------|
| Source | 2,963 | 569 | 1,008 | 213 | 1,000 |
| Target | 3,631 | 1,121 | 423 | 712 | 1,000 |

- DANN:
    - batch size: $(16, 32, 64)$
    - initial learning rate: $(1.0 \times 10^{-4}, 1.0 \times 10^{-3})$
    - parameter $\alpha$ for learning rate decay: $(0, 1.0)$
    - weight decay: $(1.0 \times 10^{-4}, 1.0 \times 10^{-3})$
    - dropout (encoder): $(0, 0.1, 0.2, 0.5)$
    - width of discriminator network: $(64, 128, 256)$
    - depth of discriminator network: $(2, 3)$
    - nonlinear classifier: (`True`, `False`)
    - parameter $\beta$ for adaption regularization decay: $(0.1, 1.0, 10.0)$.

- MDD:
    - batch size: $(16, 32, 64)$
    - initial learning rate: $(1.0 \times 10^{-4}, 1.0 \times 10^{-3})$
    - parameter $\alpha$ for learning rate decay: $(0, 1.0)$
    - weight decay: $(1.0 \times 10^{-4}, 1.0 \times 10^{-3})$
    - dropout (encoder): $(0, 0.1, 0.2, 0.5)$
    - nonlinear classifier: (`True`, `False`)
    - maximum norm value for classifier weights: $(0.5, 1.0, 2.0)$
    - parameter $\beta$ for adaption regularization decay: $(0.1, 1.0, 10.0)$.

## A.5 Entity Classification With Multiple Regions of Interest as PI

In the entity classification experiment, we train all models for at most 50 epochs. If the validation AUC does not improve for 10 subsequent epochs, we stop the training earlier. No pretrained weights are used in this experiment since we find that the task is too easy to solve with pretrained weights. For DALUPI and LUPI, we use the end-to-end solution based on Faster R-CNN (see Section A.2). We use the default anchor sizes for each of the feature maps (32, 64, 128, 256, 512), and for each anchor size we use the default aspect ratios (0.5, 1.0, 2.0). We use the binary cross entropy loss for SL-S, SL-T, DANN, and MDD.

We use the 2017 version of the MS-COCO dataset (Lin et al., 2014). As decribed in Section 4.3, we extract indoor images by sorting out images from the super categories "indoor" and "appliance" that also contain at least one of the entity classes. Outdoor images are extracted in the same way using the super categories "vehicle" and "outdoor". Images that match both domains (for example an indoor image with a toy car) are removed, as are any gray-scale images. We also include 1,000 negative examples, i.e., images with none of the entities present, in both domains. In total, there are 5,231 images in the source domain and 5,719 images in the target domain. From these pools, we randomly sample 3,000, 1,000, and 1,000 images for training, validation, and testing, respectively. In Table 5 we describe the label distribution in both domains. All images are resized to $320 \times 320$.

### A.5.1 Hyperparameters

We randomly choose hyperparameters from the following predefined sets of values. For information about the specific parameters in LUPI and DALUPI, we refer to the paper by Ren et al. (2016). Here, RoI and NMS refer to region of interest and non-maximum suppression, respectively.

- SL-S and SL-T:

    - batch size: $(16, 32, 64)$
    - initial learning rate: $(1.0 \times 10^{-4}, 1.0 \times 10^{-3})$
    - step size $n$ for learning rate decay: $(15, 30, 100)$
    - weight decay: $(1.0 \times 10^{-4}, 1.0 \times 10^{-3})$
    - dropout (encoder): $(0, 0.1, 0.2, 0.5)$
    - nonlinear classifier: (`True`, `False`).

- DANN:

    - batch size: $(16, 32, 64)$
    - initial learning rate: $(1.0 \times 10^{-4}, 1.0 \times 10^{-3})$
    - parameter $\alpha$ for learning rate decay: $(0, 1.0)$
    - weight decay: $(1.0 \times 10^{-4}, 1.0 \times 10^{-3})$
    - dropout (encoder): $(0, 0.1, 0.2, 0.5)$
    - width of discriminator network: $(64, 128, 256)$
    - depth of discriminator network: $(2, 3)$
    - nonlinear classifier: (`True`, `False`)
    - parameter $\beta$ for adaption regularization decay: $(0.1, 1.0, 10.0)$.

- MDD:

    - batch size: $(16, 32, 64)$
    - initial learning rate: $(1.0 \times 10^{-4}, 1.0 \times 10^{-3})$
    - parameter $\alpha$ for learning rate decay: $(0, 1.0)$
    - weight decay: $(1.0 \times 10^{-4}, 1.0 \times 10^{-3})$
    - dropout (encoder): $(0, 0.1, 0.2, 0.5)$
    - nonlinear classifier: (`True`, `False`)
    - maximum norm value for classifier weights: $(0.5, 1.0, 2.0)$
    - parameter $\beta$ for adaption regularization decay: $(0.1, 1.0, 10.0)$.

- LUPI and DALUPI:

    - batch size: $(16, 32, 64)$
    - learning rate: $(1.0 \times 10^{-4}, 1.0 \times 10^{-3})$
    - step size $n$ for learning rate decay: $(15, 30, 100)$
    - weight decay: $(1.0 \times 10^{-4}, 1.0 \times 10^{-3})$
    - IoU foreground threshold (RPN): $(0.6, 0.7, 0.8, 0.9)$
    - IoU background threshold (RPN): $(0.2, 0.3, 0.4)$
    - batchsize per image (RPN): $(32, 64, 128, 256)$
    - fraction of positive samples (RPN): $(0.4, 0.5, 0.6, 0.7)$
    - NMS threshold (RPN): $(0.6, 0.7, 0.8)$
    - RoI pooling output size (Fast R-CNN): $(5, 7, 9)$
    - IoU foreground threshold (Fast R-CNN): $(0.5, 0.6)$
    - IoU background threshold (Fast R-CNN): $(0.4, 0.5)$
    - batchsize per image (Fast R-CNN): $(16, 32, 64, 128)$
    - fraction of positive samples (Fast R-CNN): $(0.2, 0.25, 0.3)$
    - NMS threshold (Fast R-CNN): $(0.4, 0.5, 0.6)$
    - detections per image (Fast R-CNN): $(25, 50, 75, 100)$.

Table 6: Marginal distribution of labels of images and bounding boxes in the source and target domain, respectively, for the chest X-ray classification experiment. ATL=Atelectasis; CM=Cardiomegaly; PE=Effusion; NF=No Finding.

| Data | ATL | CM | PE | NF |
|------|-----|-----|-----|-----|
| $x \sim \mathcal{S}$ | 11,559 | 2,776 | 13,317 | 60,361 |
| $w \sim \mathcal{S}$ | 180 | 146 | 153 | - |
| $\tilde{x} \sim \mathcal{T}$ | 14,278 | 20,466 | 74,195 | 16,996 |
| $\tilde{w} \sim \mathcal{T}$ | 75 | 66 | 64 | - |

### A.6 X-ray Classification With Multiple Regions of Interest as PI

In the X-ray classification experiment, we train all models for at most 50 epochs, using pre-trained weights in the ResNet architecture of each model. If the validation AUC does not improve for 10 subsequent epochs, we stop the training earlier. We then fine-tune all models, except DANN and MDD, for up to 20 additional epochs. The number of encoder layers that are fine-tuned is a hyperparameter for which we consider different values. We start the training with weights pretrained on ImageNet. For DALUPI, we use the end-to-end solution based on Faster R-CNN (see Section A.2). We use the default anchor sizes for each of the feature maps (32, 64, 128, 256, 512), and for each anchor size we use the default aspect ratios (0.5, 1.0, 2.0). We use the binary cross entropy loss for SL-S, SL-T, DANN, and MDD.

In total, there are 83,519 (457) and 120,435 (118) images (annotated images) in the source and target domain, respectively. The distributions of labels and bounding box annotations are provided in Table 6. Here, "NF" refers to images with no confirmed findings. In the annotated images, there are 180/146/153 and 75/66/64 examples of ATL/CM/PE in each domain respectively. Validation and test sets are sampled from non-annotated images and contain 10,000 samples each. All annotated images are reserved for training. We merge the default training and validation datasets before splitting the data and resize all images to $320 \times 320$. For the source dataset (ChestX-ray8), the bounding boxes can be found together with the dataset. The target segmentations can be found here: `https://stanfordaimi.azurewebsites.net/datasets/23c56a0d-15de-405b-87c8-99c30138950c`.

#### A.6.1 Hyperparameters

We choose hyperparameters randomly from the following predefined sets of values. For information about the specific parameters in DALUPI, we refer to the paper by Ren et al. (2016). RoI and NMS refer to region of interest and non-maximum suppression, respectively.

- SL-S and SL-T:
    - batch size: (16, 32, 64)
    - learning rate: $(1.0 \times 10^{-4}, 1.0 \times 10^{-3})$
    - weight decay: $(1.0 \times 10^{-4}, 1.0 \times 10^{-3})$
    - dropout (encoder): (0, 0.1, 0.2, 0.5)
    - nonlinear classifier: (`True`, `False`)
    - number of layers to fine-tune: (3, 4, 5)
    - learning rate (fine-tuning): $(1.0 \times 10^{-5}, 1.0 \times 10^{-4})$.

- DANN:
    - batch size: $(16, 32, 64)$
    - initial learning rate: $(1.0 \times 10^{-4}, 1.0 \times 10^{-3})$
    - parameter $\alpha$ for learning rate decay: (0, 1.0)
    - weight decay: $(1.0 \times 10^{-4}, 1.0 \times 10^{-3})$

- number of trainable layers (encoder): (1, 2, 3, 4, 5)
- dropout (encoder): $(0, 0.1, 0.2, 0.5)$
- width of discriminator network: $(64, 128, 256)$
- depth of discriminator network: $(2, 3)$
- nonlinear classifier: (`True`, `False`)
- parameter $\beta$ for adaption regularization decay: (0.1, 1.0, 10.0).

- MDD:

  - batch size: $(16, 32, 64)$
  - initial learning rate: $(1.0 \times 10^{-4}, 1.0 \times 10^{-3})$
  - parameter $\alpha$ for learning rate decay: (0, 1.0)
  - weight decay: $(1.0 \times 10^{-4}, 1.0 \times 10^{-3})$
  - number of trainable layers (encoder): (1, 2, 3, 4, 5)
  - dropout (encoder): $(0, 0.1, 0.2, 0.5)$
  - nonlinear classifier: (`True`, `False`)
  - maximum norm value for classifier weights: (0.5, 1.0, 2.0)
  - parameter $\beta$ for adaption regularization decay: (0.1, 1.0, 10.0).

- DALUPI:

  - batch size: $(16, 32, 64)$
  - learning rate: $(1.0 \times 10^{-4})$
  - weight decay: $(1.0 \times 10^{-4}, 1.0 \times 10^{-3})$
  - IoU foreground threshold (RPN): (0.6, 0.7, 0.8, 0.9)
  - IoU background threshold (RPN): (0.2, 0.3, 0.4)
  - batchsize per image (RPN): (32, 64, 128, 256)
  - fraction of positive samples (RPN): (0.4, 0.5, 0.6, 0.7)
  - NMS threshold (RPN): (0.6, 0.7, 0.8)
  - RoI pooling output size (Fast R-CNN): (5, 7, 9)
  - IoU foreground threshold (Fast R-CNN): (0.5, 0.6)
  - IoU background threshold (Fast R-CNN): (0.4, 0.5)
  - batchsize per image (Fast R-CNN): (16, 32, 64, 128)
  - fraction of positive samples (Fast R-CNN): (0.2, 0.25, 0.3)
  - NMS threshold (Fast R-CNN): (0.4, 0.5, 0.6)
  - detections per image (Fast R-CNN): (25, 50, 75, 100)
  - learning rate (fine-tuning): $(1.0 \times 10^{-5}, 1.0 \times 10^{-4})$
  - number of layers to fine-tune: (3, 4, 5).

## B   Additional Results

In Figure 8a and 8b, we show some example images from the digit classification task with associated saliency maps from the source-only model for different values of the skew parameter $\epsilon$. We can see that for a lower value of epsilon the SL-S model activations seem concentrated on the area with the digit, while when the correlation with the background is large the model activations are more spread out.

In Figure 9, we show the *average* AUC when additional training data of up to 30,000 samples are added in the chest X-ray experiment. We see that, once given access to a much larger amount of labeled samples, SL-S and DALUPI perform comparably in the target domain.

In Figure 10, we show AUC for the pathology CM when additional training data *without* bounding box annotations are added. We see that SL-S catches up to the performance of DALUPI when a large amount of labeled examples are provided. These results indicate that identifiability is not the primary obstacle for adaptation, and that PI improves sample efficiency.

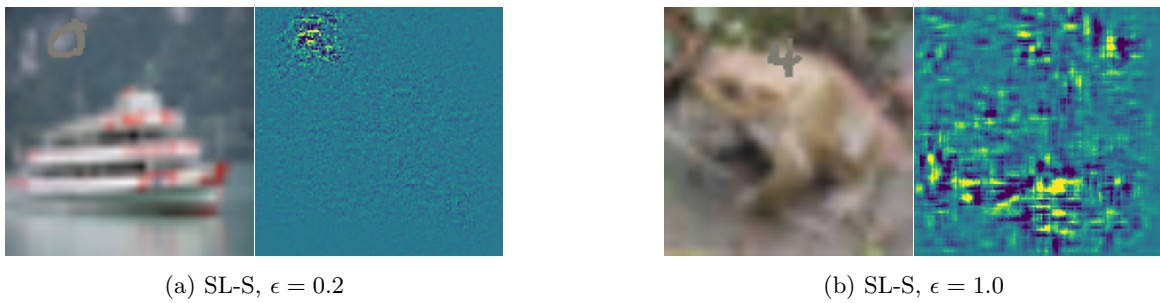

(a) SL-S, $\epsilon = 0.2$        (b) SL-S, $\epsilon = 1.0$

Figure 8: Example images (top) and saliency maps (bottom) from SL-S when trained with source skew $\epsilon = 0.2$ (a) and $\epsilon = 1$ (b).

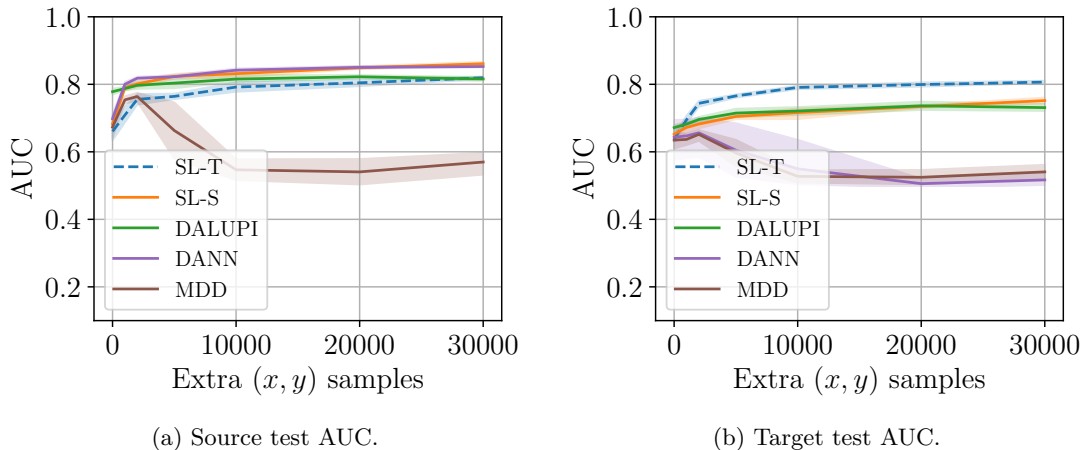

(a) Source test AUC.        (b) Target test AUC.

Figure 9: Classification of chest X-ray images. Model performance on source (a) and target (b) domains. The AUC is averaged over the three pathologies: ATL, CM and PE. The 95 % confidence intervals are computed using bootstrapping the results over five seeds.

## C    Proof of Proposition 1

**Proposition.** Let Assumptions 1 and 2 be satisfied w.r.t. $W$ (not necessarily w.r.t. $X$) and let Assumption 3 hold as stated. Then, the target risk $R_{\mathcal{T}}$ is identified for hypotheses $h : \mathcal{X} \to \mathcal{Y}$,

$$R_{\mathcal{T}}(h) = \int_x \mathcal{T}(x) \int_w \mathcal{T}(w \mid x) \int_y \mathcal{S}(y \mid w) L(h(x), y) \mathrm{d}y \mathrm{d}w \mathrm{d}x \ .$$

and, for $L$ the squared loss, a minimizer of $R_{\mathcal{T}}$ is $h_{\mathcal{T}}^*(x) = \int_w \mathcal{T}(w \mid x) \mathbb{E}_{\mathcal{S}}[Y \mid w] \mathrm{d}w$ .

*Proof.* By definition, $R_{\mathcal{T}}(h) = \int_{x,y} \mathcal{T}(x,y) L(h(x), y) \mathrm{d}y \mathrm{d}x$. We marginalize over $W$ to get

$$\begin{aligned}
\mathcal{T}(x,y) &= \mathcal{T}(x) \mathbb{E}_{\mathcal{T}(W \mid x)} \left[ \mathcal{T}(y \mid W, x) \mid x \right]] \\
&= \mathcal{T}(x) \mathbb{E}_{\mathcal{T}(W \mid x)} [ \mathcal{T}(y \mid W) \mid x ] \\
&= \mathcal{T}(x) \int_{w : \mathcal{T}(w) > 0} \mathcal{T}(w \mid x) \mathcal{S}(y \mid w) \mathrm{d}w \\
&= \mathcal{T}(x) \int_{w : \mathcal{S}(w) > 0} \mathcal{T}(w \mid x) \mathcal{S}(y \mid w) \mathrm{d}w \ .
\end{aligned}$$

where the second equality follows by sufficiency and the third by covariate shift and overlap in $W$. $\mathcal{T}(x), \mathcal{T}(w \mid x)$ and $\mathcal{S}(y \mid w)$ are observable through training samples. That $h_{\mathcal{T}}^*$ is a minimizer follows from the first-order

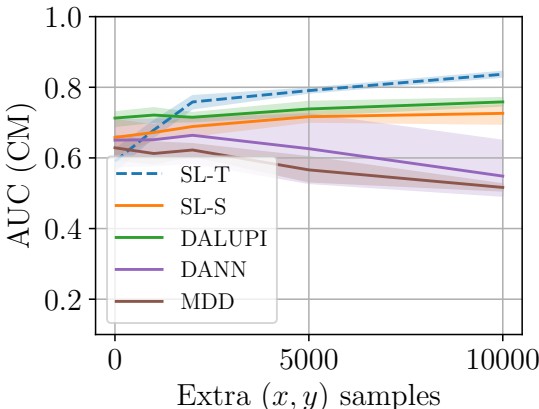

Figure 10: Test AUC for CM in $\mathcal{T}$. DALUPI outperforms the other models when no extra $(x,y)$ samples are provided. Adding examples without bounding box annotations improves the performance of SL-S and SL-T, eventually causing the latter to surpass DALUPI.

condition of setting the derivative of the risk with respect to $h$ to 0. This strategy yields the well-known result that

$$h^*_{\mathcal{T}} = \arg\min_h \mathbb{E}_{\mathcal{T}}[(h(X) - Y^2)] = \mathbb{E}_{\mathcal{T}}[Y \mid X] \; .$$

By definition and the previous result, we have that

$$\mathbb{E}_{\mathcal{T}}[Y \mid X = x] = \int_y y \frac{\mathcal{T}(x,y)}{\mathcal{T}(x)} \mathrm{d}y$$

$$= \int_y \int_{w:\mathcal{S}(w)>0} \mathcal{T}(w \mid x)\mathcal{S}(y \mid w)y\mathrm{d}w\mathrm{d}y$$

$$= \int_w \mathcal{T}(w \mid x)\mathbb{E}_{\mathcal{S}}[Y \mid x]\mathrm{d}w$$

and we have the result. $\qquad\qquad\square$

## D  Proof of Proposition 2

**Proposition 2.**  *Assume that $\mathcal{G}$ comprises M-Lipschitz mappings from the privileged information space $\mathcal{W} \subseteq \mathbb{R}^{d_W}$ to $\mathcal{Y}$. Further, assume that both the ground truth privileged information $W$ and label $Y$ are deterministic in $X$ and $W$ respectively. Let $\rho$ be the domain density ratio of $W$ and let Assumptions 1–3 (Covariate shift, Overlap and Sufficiency) hold w.r.t. $W$. Further, let the loss $L$ be uniformly bounded by some constant $B$ and let $d$ and $d'$ be the pseudo-dimensions of $\mathcal{G}$ and $\mathcal{F}$ respectively. Assume that there are $n$ observations from the source (labeled) domain and $m$ from the target (unlabeled) domain. Then, with $L$ the squared Euclidean distance, for any $h = h \circ f \in \mathcal{G} \times \mathcal{F}$, w.p. at least $1 - \delta$,*

$$\frac{R_{\mathcal{T}}(h)}{2} \leq \hat{R}^{Y,\rho}_{\mathcal{S}}(g) + M^2 \hat{R}^W_{\mathcal{T}}(f)$$

$$+ 2^{5/4}\sqrt{d_2(\mathcal{T}\|\mathcal{S})} \sqrt[3/8]{\frac{d\log\frac{2me}{d} + \log\frac{4}{\delta}}{m}}$$

$$+ d_{\mathcal{W}}BM^2\left(\sqrt{\frac{2d'\log\frac{en}{d'}}{n}} + \sqrt{\frac{\log\frac{d_{\mathcal{W}}}{\delta}}{2n}}\right).$$

*Proof.* Decomposing the risk of $h \circ \phi$ , we get

$$
\begin{aligned}
R_{\mathcal{T}}(h) &= \mathbb{E}_{\mathcal{T}}[(g(f(X)) - Y)^2] \\
&\le 2\mathbb{E}_{\mathcal{T}}[(g(W) - Y)^2 + (g(f(X)) - g(W))^2] \\
&\le 2\mathbb{E}_{\mathcal{T}}[(g(W) - Y)^2 + M^2\|f(X)) - g(W)\|^2] \\
&\le 2\mathbb{E}_{\mathcal{T}}[(g(W) - Y)^2] + 2M^2\mathbb{E}_{\mathcal{T}}[\|(f(X) - W)\|^2] \\
&= 2R_{\mathcal{T}}^Y(g) + 2M^2 R_{\mathcal{T}}^W(f) = 2\underbrace{R_S^{Y,\rho}(g)}_{(I)} + 2M^2\underbrace{R_{\mathcal{T}}^W(f)}_{(II)}.
\end{aligned}
$$

The first inequality follows from the relaxed triangle inequality, the second inequality from the Lipschitz property and the third equality from Overlap and Covariate shift. We will bound these quantities separately starting with $(I)$.

We assume that the pseudo-dimension of $\mathcal{G}$, $d$ is bounded. Further, we assume that the second moment of the density ratios, equal to the Rényi divergence $d_2(\mathcal{T}\|\mathcal{S}) = \Sigma_{w \in cG}\mathcal{T}(w)\frac{\mathcal{T}(w)}{\mathcal{S}(w)}$ are bounded and that the density ratios are non-zero for all $w \in \mathcal{G}$. Let $D_1 = \{w_i, y_i\}_{i=0}^m$ be a dataset drawn i.i.d from the source domain. Then by application of Theorem 3 from Cortes et al. (2010) we obtain with probability $1 - \delta$ over the choice of $D_1$,

$$
(I) = R_S^{Y,\rho}(g) \le \hat{R}_S^{Y,\rho}(g) + 2^{5/4}\sqrt{d_2(\mathcal{T}\|\mathcal{S})}\sqrt[3/8]{\frac{d\log\frac{2me}{d} + \log\frac{4}{\delta}}{m}}
$$

Now for $(II)$ we treat each component of $w \in \mathcal{W}$ as a regression problem independent from all the others. So we can therefore write the risk as the sum of the individual component risks

$$
R_{\mathcal{T}}^W(f) = \Sigma_{i=1}^{d_{\mathcal{W}}} R_{\mathcal{T},i}^W(f)
$$

Let the pseudo-dimension of $\mathcal{F}$ be denoted $d$, $D_2 = \{x_i, w_i\}_{i=0}^n$ be a dataset drawn i.i.d from the target domain. Then, using theorem 11.8 from Mohri et al. (2018) we have that for any $\delta > 0$, with probability at least $1 - \delta$ over the choice of $D_2$, the following inequality holds for all hypotheses $f \in \mathcal{F}$ for each component risk

$$
R_{\mathcal{T},i}^W(f) \le \hat{R}_{\mathcal{T},i}^W(f) + B\left(\sqrt{\frac{2d'\log\frac{en}{d'}}{n}} + \sqrt{\frac{\log\frac{1}{\delta}}{2n}}\right)
$$

We then simply make all the bounds hold simultaneously by applying the union bound and having it so that each bound must hold with probability $1 - \frac{\delta}{d_{\mathcal{W}}}$ which results in

$$
\begin{aligned}
R_{\mathcal{T}}^W(f) = \Sigma_{i=1}^{d_{\mathcal{W}}} R_{\mathcal{T},i}^W(f) &\le \Sigma_{i=1}^{d_{\mathcal{W}}} \hat{R}_{\mathcal{T},i}^W(f) + \Sigma_{i=1}^{d_{\mathcal{W}}} B\left(\sqrt{\frac{2d'\log\frac{en}{d'}}{n}} + \sqrt{\frac{\log\frac{d_{\mathcal{W}}}{\delta}}{2n}}\right) \\
&= \hat{R}_{\mathcal{T}}^W(f) + d_{\mathcal{W}}B\left(\sqrt{\frac{2d'\log\frac{en}{d'}}{n}} + \sqrt{\frac{\log\frac{d_{\mathcal{W}}}{\delta}}{2n}}\right)
\end{aligned}
$$

Combination of these two results then yield the proposition statement.

Consistency follows as $Y$ is a deterministic function of $W$ and $W$ is a deterministic fundtion of $X$ and both $\mathcal{H}$ and $\mathcal{F}$ are well-specified. Thus both empirical risks and sample complexity terms will converge to 0 in the limit of infinite samples. $\square$

The parts of the bound shown above can be described as falling into three main categories: Empirical risk(s), domain shift and sample complexity components. A central term that figures both in the weighted empirical

risk and the Rényi divergence is the density ratio $\frac{\mathcal{T}(w)}{\mathcal{S}(w)}$. Therefore, the size of the bound is governed at least in part based on the proximity in $\mathcal{W}$-space the source and target domains are. This is similar to other importance weighting bounds, however, since the experiment designer may choose the form of PI this can be more well-behaved than the density ratio in the input space.

# E    Proof Sketch for PAC-Bayes Bound

We will here detail a proof sketch for a PAC-Bayes version of the bound we propose in the main text. For the purposes of this bound we will consider the quantity $\mathbb{E}_{h\sim\psi}R_{\mathcal{T}}(h)$, where $\psi$ is a posterior distribution over classifiers $h\sim\psi$. As we are basing the bound on the two-step methodology where we train two different classifiers on separate datasets we assume that we can obtain the posteriors over the component functions separately and independently i.e. $h = f \circ g \sim \psi = \psi_f \times \psi_g$, where $f \sim \psi_f$ and $g \sim \psi_g$. Let the assumptions from proposition 2 hold here. Similar to the previous section we decompose the risk into two parts

$$
\begin{aligned}
\mathbb{E}_{h\sim\psi}R_{\mathcal{T}}(h) &= \mathbb{E}_{h\sim\psi}\mathbb{E}_{\mathcal{T}}[(g(f(X)) - Y)^2] \\
&= \mathbb{E}_{h\sim\psi}[2R_{\mathcal{T}}^Y(g) + 2M^2 R_{\mathcal{T}}^W(f)] = 2\mathbb{E}_{h\sim\psi}\underbrace{R_{\mathcal{T}}^Y(g)}_{(I)} + 2M^2\mathbb{E}_{h\sim\psi}\underbrace{R_{\mathcal{T}}^W(f)}_{(II)}.
\end{aligned}
$$

We note that since we now have expectations over the composite function $h$ on expressions which depend on only one of the components we can, for example, write the following:

$$
\mathbb{E}_{h\sim\psi}R_{\mathcal{T}}^Y(g) = \mathbb{E}_{g\sim\psi_g}R_{\mathcal{T}}^Y(g)
$$

This holds as we assume that f and g are not dependent on each other. Therefore, we can just marginalize out the part which is not in use. From this point we can use some of the available bounds from the literature to estimate the resulting part e.g. Corollary 1 from Breitholtz & Johansson (2022). Application of this result yields the following bound on the first term

$$
\mathbb{E}_{g\sim\psi_g}R_{\mathcal{T}}^Y(g) \le \frac{1}{\gamma}\mathbb{E}_{g\sim\psi_g}R_{\mathcal{S}}^{\hat{Y},\rho}(g) + \beta_\infty\frac{\mathrm{KL}(\psi_g\|\pi_g) + \ln(\frac{1}{\delta})}{2\gamma(1-\gamma)m}\ .
$$

Thereafter we can use another bound from the literature to estimate the second term, e.g. Theorem 6 from Germain et al. (2020). Using this we obtain the following:

$$
\mathbb{E}_{f\sim\psi_f}R_{\mathcal{T}}^W(f) \le \frac{\alpha}{1 - e^{-\alpha}}\left(\mathbb{E}_{f\sim\psi_f}\hat{R}_{\mathcal{T}}^W(f) + \frac{\mathrm{KL}(\psi_f\|\pi_f) + \ln(\frac{1}{\delta})}{n\alpha}\right)\ .
$$

Then a bound can be constructed by combining these two results using a union bound argument.

$$
\begin{aligned}
\mathbb{E}_{h\sim\psi}R_{\mathcal{T}}(h) \le{}& \frac{2}{\gamma}\mathbb{E}_{g\sim\psi_g}R_{\mathcal{S}}^{\hat{Y},\rho}(g) + \beta_\infty\frac{\mathrm{KL}(\psi_g\|\pi_g) + \ln(\frac{2}{\delta})}{2\gamma(1-\gamma)m} \\
&+ \frac{2M^2\alpha}{1 - e^{-\alpha}}\left(\mathbb{E}_{f\sim\psi_f}\hat{R}_{\mathcal{T}}^W(f) + \frac{\mathrm{KL}(\psi_f\|\pi_f) + \ln(\frac{2}{\delta})}{n\alpha}\right)
\end{aligned}
$$

# F    A Bound on the Target Risk Without Suffiency

The sufficiency assumption is used to replace $\mathcal{T}(y \mid x)$ with $\mathcal{T}(y \mid w)$ in the proof of Proposition 1. If sufficiency is violated but it is plausible that the degree of insufficiency is comparable across domains, we can still obtain a bound on the target risk which may be estimated from observed quantities. One way to formalize such an assumption is that there is some $\gamma \ge 1$, for which

$$
\sup_{x\in\mathcal{T}(x|w)} \mathcal{T}(y \mid w, x)/\mathcal{T}(y \mid w) \le \gamma \sup_{x\in\mathcal{S}(x|w)} \mathcal{S}(y \mid w, x)/\mathcal{S}(y \mid w) \tag{5}
$$

This may be viewed as a relaxation of suffiency. If Assumption 3 holds, both left-hand and right-hand sides of the inequality are 1. Under Equation 5, with $\Delta_\gamma(w, y)$ equal to the right-hand side the inequality,

$$R_\mathcal{T}(h) \leq \int_x \mathcal{T}(x) \int_w \mathcal{T}(w \mid x) \int_y \Delta_\gamma(w, y)\mathcal{S}(y \mid w)L(h(x), y)\mathrm{d}y\mathrm{d}w\mathrm{d}x \ .$$

However, the added assumption is not verifiable statistically.

