# OpenReview forum: "Unsupervised Domain Adaptation by Learning Using Privileged Information"
_TMLR — Accepted by TMLR_

### Review · Reviewer_hqvf · 2024-05-28

**Summary Of Contributions:**

The paper considers the unsupervised domain adaptation (DA) problem with the privileged information. This means that the learner has access to data triplets $(x,w,y)\sim S$ from the source data distributions and has to build a predictor $h(x)$ that will be applied to some different but related target distribution $x\sim T$ (actually, $(x,w,y)\sim T$ but $w,y$ are not observed during testing). The predictor is expected to perform well on $T$.

The authors of the paper claim that standard theoretically-justified approaches to DA which use only $(x,y)$ usually require non-realistic assumptions, e.g., on $S, T$, which do not generally hold in practice. The main message of the paper is that the usage of privileged information $w$ (which is, of course, not always available), can help to alleviate this issue. The authors provide a two-stage algorithm (train prediction for priviledged information $w=f(x)$ and predictor for the label $y=g(w)$ on the train data from $S$; use $h(x)=g(f(x))$ to predict on $T$) to perform DA using the privileged information and analyze their approach from the empirical risk perspective. From the practical point of view, the authors do several computational experiments (outperform some DA baselines in several tasks:  celeba, digits, entity and X-ray classification) and also demonstrate a particular (though, synthetic) experiment which evidently demonstrates the case when using the privileged information yields superior performance compared to standard approach which does not use it.

Since I am not a deep expert in the field of unsupervised domain adaptation, it is possible that in my review I missed some important aspect as I am not very familiar with deep aspects of existing research/baselines in the subfield.

**Audience:**

Yes

**Broader Impact Concerns:**

No concerns

**Claims And Evidence:**

Yes

**Requested Changes:**

Since I am not a deep expert in the field of domain adaptation, I can not for sure say that the baselines considered in the paper are state-of-the-art and setups are truly ok. Therefore, my review is mostly about logical and presentational aspects of the paper.

I would like the authors to Improve the presentation/explanation of their theoretical results as well as improve the mathematical rigor of their claims and proofs, see my comments in the "weaknesses" section above.

**Strengths And Weaknesses:**

**Strength**
- The overall proposed idea of exploiting additional privileged information for DA seems rather natural, reasonable and inspiring.
- The overall clarity is mostly ok (especially the main message of the paper and its explanations).
- The paper provides theoretical guarantees of the provided approach's performance. This is a positive contrast aspect distinguishing the work from many modern domain adaptation papers which are purely heuristical.
- Illustrative evaluation on 3 problems with different types of privileges information is conducted (binary attributes, single region, multiple regions). The method, in general, outperforms the baselines or performs comparably.

**Weaknesses**
- While the theoretical result (risk bound, proposition 2) looks interesting, it is not very well explained. The authors should provide a deeper discussion of what each component means and how it affects the bound. I especially talk about the density ratio $p(w)=T(w)/S(w)$ which seems to be the most unclear component here. At the same time, it is one of the most important as it explains how far $T$ is from $S$. The phrase “we may use density estimation” confuses.
- Some assumptions of the paper which are needed to derive the theoretical results look too unrealistic. For example, I think that “Suppose that $W$ and $Y$ are deterministic in $X$ and $W$, respectively” is a very unrealistic assumption. Assumption 3 is also not very practical. Given that, the derived theoretical results are not truly inspiring as they are mostly straightforward combinations of existing statistical learning theory results and decompositions.
- I would say that the paper is not sufficiently mathematically rigorous. For example, in Proposition 1 there are sums over x,w,y while it should contain integrals over the respective data distributions (or expectations). This is somewhat strange (taking into account that sometimes the authors use the notation with expectations themself). In equation (2) there are the same loss functions $L$ for learning privileged information and prediction labels which seems to be not correct (in general, they are expected to be different, right?).
- Synthetic experiment in section 4.1 is a little bit hard to parse. It would be nice to have a table or smth like it briefly summarizing how and where the covariate shift happens in the considered case (it is explained in the text but I do not completely understand).

---

> ### Author Response · Authors · 2024-07-05
> **Response to Reviewer hqvf**
>
> We would like to thank Reviewer hqvf for their insightful comments and questions. We respond to each point in turn below.
>
>
> **R: While the theoretical result (risk bound, proposition 2) looks interesting, it is not very well explained. The authors should provide a deeper discussion of what each component means and how it affects the bound. I especially talk about the density ratio $p(w)=T(w)/S(w)$ which seems to be the most unclear component here. At the same time, it is one of the most important as it explains how far $T$ is from $S$. The phrase "we may use density estimation" confuses.**
>
> A: We thank the reviewer for pointing out this ambiguity. We have now clarified the role of the density ratio $\rho$ in the proof sketch of Proposition 1. We have also added additional discussion of the bound in Appendix D.
>
>
> **R: Some assumptions of the paper which are needed to derive the theoretical results look too unrealistic. For example, I think that "Suppose that $W$ and $Y$ are deterministic in $X$ and $W$, respectively" is a very unrealistic assumption. Assumption 3 is also not very practical. Given that, the derived theoretical results are not truly inspiring as they are mostly straightforward combinations of existing statistical learning theory results and decompositions.**
>
> A: We agree that some of the assumptions may be cumbersome. However, the assumptions that $W$ and $Y$ are deterministic are not necessarily unrealistic in all cases. There are examples where $W$ is a deterministic function of the input such as a crop of an image like we have in our paper. Then, the label is also deterministic of the cropped portion of the image, see Figure 2 for an illustration where the relevant pixels for the label are within the cropped portion.
>
>
> **R: I would say that the paper is not sufficiently mathematically rigorous. For example, in Proposition 1 there are sums over $x,w,y$ while it should contain integrals over the respective data distributions (or expectations). This is somewhat strange (taking into account that sometimes the authors use the notation with expectations themself). In equation (2) there are the same loss functions $L$ for learning privileged information and prediction labels which seems to be not correct (in general, they are expected to be different, right?).**
>
> A: While an expectation can be written in terms of a sum or an integral (depending on whether the distribution is discrete or continuous), we see where there might be some confusion. We have made changes to Proposition 1, changing sums to integrals and adding expectations for increased clarity. Regarding the loss functions the reviewer is correct, they may of course be different in general. We have added a note and some notation in the revision clarifying this.
>
>
> **R: Synthetic experiment in section 4.1 is a little bit hard to parse. It would be nice to have a table or smth like it briefly summarizing how and where the covariate shift happens in the considered case (it is explained in the text but I do not completely understand).**
>
> A: The covariate shift we point out is that for the case where there are no attributes present. In this case, the majority label changes from one domain to the other, violating covariate shift w.r.t. $W$. We have clarified this in Section 4.1 paragraph §3 of the updated manuscript. This experiment is adapted from the work by Xie et al., and we have clarified that the reader can find more information in their paper.
>
> **R: This means that the learner has access to data triplets $(x,w,y)\sim S$ from the source data distributions and has to build a predictor $h(x)$ that will be applied to some different but related target distribution $x\sim T$ (actually, $(x,w,y)\sim T$ but $w,y$ are not observed during testing).**
>
> A: It seems like the reviewer suggests that we have access to target labels during training. As stated in, e.g., the second paragraph of Section 2.1, this is not the case. We only assume access to the image $X$ and privileged information $W$ from the target domain during training.

---

> > ### Comment · Reviewer_hqvf · 2024-07-08
> > **Minor comment**
> >
> > Thanks for revising the paper, but I think that it is still a little bit messy in terms of notation. For example, in the paper (in particular, in proposition 1), you sometimes use capital letters (X,W) while sometimes you use small letters (x,w). This does not look very consistent (you also use both T(X) and T(x) in the paper). In the proof of proposition 1 there is “dw” forgotten in the integral. In the formulation of the proposition, it seems like your risk is a product of the integrals rather than the triple integral. Could you please double check again your entire paper and proofs for mistakes and misprints in notation or formulas and, if possible, consider unifying the notation somehow?

---

> > > ### Author Response · Authors · 2024-07-10
> > > **Re:**
> > >
> > > Thank you for your continued engagement with our work. It is much appreciated!
> > >
> > > We aim to follow the convention of writing random variables with capital roman letters (e.g., in expectations) and their observations with lower-case roman letters (e.g., the integration variable). That is, both $T(X)$ and $T(x)$ should appear, depending on context. We use the common shorthand $\mathcal{T}(x)$ for $\mathcal{T}(X=x)$ to mean the density of the distribution $\mathcal{T}$ of the random variable $X$, evaluated at $x$. We thank you for pointing out inconsistencies in us following this convention. We have added a note after equation (1) to clarify what we mean by $\mathbb{E}_{p(X)}[f(X)]$ for some density $p$ and function $f$.
> > >
> > > We have updated the manuscript again to fix the issues surrounding Proposition 1. Since it uses several nested conditional expectations, we have chosen to keep (but clarify) their integral forms.

---

### Review · Reviewer_PbCi · 2024-06-04

**Summary Of Contributions:**

The paper presents the notion of unsupervised domain adaptation with privileged information (PI). This means to achieve adaptation of machine learning models trained for a given task over similar but not identical domains. We can think of this as having measurements taken by some medical imaging devices, which are analyzed by different healthcare experts for a given of identifying a structure in the image, the possible structures being the same over all cases. The task stays the same, but the measurements and expert interpretation may vary. Over this domain adaptation, the paper is proposing an approach to make use of privileged information, that is, side information elements of the data that can be used to improve training of the models. The paper is developing the idea, with some theoretical developments, a technical proposal, and experiments demonstrating the proper working of the approach.

The main contribution of the paper is to develop the use of PI from the target domain, not just the source domain, as it has been proposed previously (i.e., LUPI). The developments and experiments support the proposal that using PI for target domain is helpful in the context of domain adaptation.

**Audience:**

Yes

**Broader Impact Concerns:**

The contribution is rather technical and doesn’t raise any specific ethical issues.

**Claims And Evidence:**

Yes

**Requested Changes:**

The limitations on the type and availability of PI for target domains should be further discussed. In particular, when the PI is not sufficient for the task, but can be helpful to improve performance as additional information the input data (X), to what extent can this be helpful?

Also, a point that is not clear to me is how to deal with missing PI from a target domain. Maybe I missed it from the paper, but it seems that in the case of having only part of the target domain data with PI, the rest not having PI, we can still proceed and use the first stage of the model to estimate the pseudo-PI of the sample. Is this correct? I think that this can be better developed, as this is an important aspect. For instance, in Fig. 4, when we show results for n_PI(T) varying between 0 and 1, I guess that the results are reported over all the dataset T from target domain, estimating PI from samples missing it.

In general, there are few details on the specific algorithms used in the main part of the paper, some more details are provided in the appendix, but even there, there are still some dots to connect by the reader. I would suggest to the authors to provide a complete, more explicit explanations on how they implemented their model in practice for the experiments conducted, both in the main paper and the appendix (for the extra details).

**Strengths And Weaknesses:**

Strengths:
- Solid theoretical arguments supporting the proposed approach
- Broad set of experiments supporting the proposal.
- Well-organized and well-written paper.

Weaknesses:
- The PI needs to be sufficient for achieving the end task, which is quite a strong assumption. The proposed two-stage algorithm is formulated that way, the second stage receiving only the W (privileged information), not the data X itself. Maybe the X itself can be included in the W, but that’s not discussed.
- The proposal is quite limited and narrow in terms of applicability, it makes sense for providing bounding boxes, but there are not many other obvious settings allowing it. The binary attributed for the celebrity photos seems more like a toy problem, and it is not clear at all that the binary attributes are sufficient for classification. The bounding box PI is relevant, but appears quite niche.
- Moreover, the fact that the PI should be available from the target domain puts an extra burden for achieving some results, and obviously provides an advantage over the task achieved. For instance, providing bounding boxes over the objects of interest requires a level of intervention that is close to completing the classification task itself (even though it is argued otherwise in the text).

---

> ### Author Response · Authors · 2024-07-05
> **Response to Reviewer PbCi**
>
> We would like to thank Reviewer PbCi for their insightful comments and questions. We respond to each point in turn below.
>
>
> **R: The PI needs to be sufficient for achieving the end task, which is quite a strong assumption. The proposed two-stage algorithm is formulated that way, the second stage receiving only the W (privileged information), not the data X itself. Maybe the X itself can be included in the W, but that’s not discussed.**
>
> A: While sufficiency is a necessary assumption for identification, it needs not always be a strong assumption. We would expect there to exist some PI that can be chosen such that it is sufficient for a given task, e.g. the region of interest which we use for the entity classification task. However, if sufficiency cannot be ensured then we would expect the overall performance to decrease somewhat, if covariate shift w.r.t. $w$ is not violated. However, we would still expect the generalization error to be comparable in this case. We have added a note about this in section 2.1.
>
> The reviewer also brings up an interesting point in passing on the data as well as the PI. This is similar to what we do in the end-to-end model. If we use a system where we pass on both $X$ and $W$ as inputs, there is a possibility that $W$ is ignored. This would then completely negate the benefits of learning with $W$. Furthermore, when sufficiency is violated, the best possible case when using $X$ as an input would be if we can achieve both the low bias of an $X$-only model and the variance reduction of a $W$-only model.
>
> **R: The proposal is quite limited and narrow in terms of applicability, it makes sense for providing bounding boxes, but there are not many other obvious settings allowing it. The binary attributed for the celebrity photos seems more like a toy problem, and it is not clear at all that the binary attributes are sufficient for classification. The bounding box PI is relevant, but appears quite niche.**
>
> A: We agree that the example with binary attributes can be seen as a toy example -- we include this experiment to compare DALUPI to In-N-Out (we did not include In-N-Out in other experiments since it is not designed to use regions of interest as privileged information). However, we disagree that the use of PI is limited to only one form.  Except when PI is provided as regions of interest, one can, e.g., imagine a situation where the task is to predict treatment, discharge, or diagnosis based on a medical image or other patient information. In this case, the domain shift is represented by two different hospitals or healthcare facilities, and privileged information could consist of binary attributes, continuous features, or clinical notes. In this example, sufficiency would be more likely to hold as much of the information related to the label is plausibly contained within the PI.
>
>
>
>
> **R: Moreover, the fact that the PI should be available from the target domain puts an extra burden for achieving some results, and obviously provides an advantage over the task achieved. For instance, providing bounding boxes over the objects of interest requires a level of intervention that is close to completing the classification task itself (even though it is argued otherwise in the text).**
>
> A: Providing a bounding box does not necessarily complete the classification task, as it is unclear whether localization information determines the class in general. We argue that providing privileged information is often easier than providing labels and we give an example with animal classification in the second paragraph of Section 2.1. Another example is anomaly detection and classification, where it typically would be easier to indicate anomalies (e.g., lung abnormalities in X-ray images) than to classify them. In practice, it is also possible that bounding boxes may be annotated with less accuracy, with the risk of violating the assumption of domain overlap in $W$. We find in experiments that DALUPI is robust to violations of assumptions of sufficiency and covariate shift.
>
>
> **R: The limitations on the type and availability of PI for target domains should be further discussed. In particular, when the PI is not sufficient for the task, but can be helpful to improve performance as additional information the input data (X), to what extent can this be helpful?**
>
> A: We thank and agree with the reviewer on this point. As we responded above, we should expect there to be some choice of PI that is sufficient for a given task. If the sufficiency is violated we would expect the overall performance to decrease somewhat, while generalization error would be similar, if covariate shift w.r.t. $w$ holds. We have added additional discussion in Section 2.1 regarding the limitations of the method when sufficiency is violated.

---

> ### Author Response · Authors · 2024-07-05
> **Response to Reviewer PbCi cont.**
>
> **R: Also, a point that is not clear to me is how to deal with missing PI from a target domain. Maybe I missed it from the paper, but it seems that in the case of having only part of the target domain data with PI, the rest not having PI, we can still proceed and use the first stage of the model to estimate the pseudo-PI of the sample. Is this correct? I think that this can be better developed, as this is an important aspect. For instance, in Fig. 4, when we show results for $n_{PI}(\mathcal{T})$ varying between 0 and 1, I guess that the results are reported over all the dataset T from target domain, estimating PI from samples missing it.**
>
> A: This is indeed an interesting point the reviewer brings up. When target PI is partly missing, there are two main ways one could approach this issue:
> 1. Use the limited amount of target PI that is available to train the first-stage estimator $\hat{f}$ and hope that the second-stage estimator $\hat{g}$ is good enough to achieve reasonable overall performance.
> 2. Similar to the reviewer's suggestion, use $\hat{f}$ to create "weak" PI labels for the inputs that are missing PI. This is similar to the work of e.g. Robinson et al. [1].
>
> We use the first approach in this work. In Section 4.3, we study the impact of limiting the number of target samples for which PI is available when training the end-to-end model. Specifically, in the right panel of Figure 5, we vary the fraction of the 3,000 target samples $(\tilde{x}, \tilde{w})$ that are available for training from 0 to 1. When about half of the target samples are available, we achieve a performance that is comparable to that of the target oracle, which has access to all target samples (including labels). This demonstrates the sample efficiency of our method. Note that we do not impute PI for samples that are missing it.
>
> The second approach could be helpful if the first-stage estimator predicts reasonable PI. However, it may bias the resulting model in unintended ways and, as such, should be undertaken with some caution. We have added a short discussion of this in Section 6.
>
> [1]: Joshua Robinson, Stefanie Jegelka, and Suvrit Sra. Strength from Weakness: Fast Learning Using Weak Supervision. In Proceedings of the 37th International Conference on Machine Learning, pp. 8127--8136. PMLR, 2020.
>
> **R: In general, there are few details on the specific algorithms used in the main part of the paper, some more details are provided in the appendix, but even there, there are still some dots to connect by the reader. I would suggest to the authors to provide a complete, more explicit explanations on how they implemented their model in practice for the experiments conducted, both in the main paper and the appendix (for the extra details).**
>
> A: We have updated the manuscript with a schematic illustration of the two-stage estimator and the end-to-end estimator to better highlight the differences between the two approaches. We have also extended the description of the end-to-end estimator in Section 3.3. In Section 4, we have specified which ResNet architecture that was used in each experiment. Moreover, the experimental details in the appendix have been updated to help the reader better understand the setup of each experiment. We will also make the code available upon acceptance. We hope that the reviewer is satisfied with these changes.

---

### Review · Reviewer_jGW2 · 2024-06-24

**Summary Of Contributions:**

This paper introduces Unsupervised Domain Adaptation by Learning Using Privileged Information (DALUPI), a novel approach to tackle the challenge of transferring knowledge from a labeled source domain to an unlabeled target domain. Traditional unsupervised domain adaptation (UDA) methods often rely on strong assumptions about domain overlap that are frequently violated in high-dimensional problems like image classification. DALUPI addresses this limitation by leveraging privileged information (PI)---additional data available only during training---to relax restrictions on input variables and improve sample efficiency. The authors provide theoretical guarantees for consistent learning without requiring distributional overlap between input domains, which is typically needed for traditional UDA methods.

To demonstrate the practical utility of DALUPI, the paper proposes learning algorithms for image classification using three types of privileged information: binary attribute vectors, single regions of interest, and multiple regions of interest. The authors evaluate their approach on four tasks, including celebrity photo classification, digit recognition, entity detection in natural images, and X-ray classification. Across these experiments, DALUPI demonstrates improved performance compared to baseline methods, especially when input domain overlap is violated and training sets are small. The results show that DALUPI can successfully adapt in scenarios where traditional UDA methods fail and exhibits increased sample efficiency compared to non-privileged learners.

**Audience:**

Yes

**Claims And Evidence:**

Yes

**Requested Changes:**

- The authors could design experiments to test how sensitive DALUPI is to violations of the sufficiency assumption. This could involve artificially degrading the quality of the privileged information to see how it affects performance. Such experiments would help understand the robustness of DALUPI in scenarios where the privileged information is not perfectly sufficient for predicting labels.

- Another thing to do would be to test DALUPI on more challenging cross-domain adaptation tasks, such as adapting between drastically different image styles (e.g., photos to sketches, or Yelp reviews to tweets). This would showcase the method's ability to handle more extreme domain shifts.'

- There is a whole line of work in NLP that calls this learning with feature feedback that you haven't discussed in related work. Several recent papers have shown how learning with feature feedback can help models have better performance vis-a-vis out of domain. Please include those in related work. It might also be beneficial t9o show your evaluate on those NLP tasks. [1, 2]

[1] Kaushik, D., Setlur, A., Hovy, E. H., & Lipton, Z. C. Explaining the Efficacy of Counterfactually Augmented Data. In International Conference on Learning Representations. 2021.
[2] Katakkar, A., Yoo, C. H., Wang, W., Lipton, Z. C., & Kaushik, D. Practical Benefits of Feature Feedback Under Distribution Shift. In Proceedings of the Fifth BlackboxNLP Workshop on Analyzing and Interpreting Neural Networks for NLP (pp. 346-355). 2022.

**Strengths And Weaknesses:**

## Strengths:

- The paper provides a theoretical framework for unsupervised domain adaptation by learning using privileged information (DALUPI). This framework relaxes the strong assumptions typically required in UDA, particularly the need for input domain overlap. This is a significant advancement in the field, as it allows for adaptation in scenarios where traditional UDA methods might fail.

- The authors conduct several experiments for image classification using three different types of privileged information, including on a real-world medical imaging application.

- The experimental results show that DALUPI models often outperform baseline methods, especially in scenarios with limited training data. In some cases, DALUPI even outperforms oracle models trained on target labels, highlighting the method's strong sample efficiency. This is a crucial advantage in many real-world scenarios where labeled data is scarce or expensive to obtain.

## Weaknesses:
- The theoretical guarantees of DALUPI rely on the assumption that the privileged information is ``sufficient'' for predicting labels. While this assumption is justified for some of the experimental tasks, it may not hold in many real-world scenarios. The paper could benefit from a more thorough discussion of when this assumption is likely to hold and how violations of it might impact performance.

- While the paper compares DALUPI to some baseline UDA methods (DANN and MDD), it doesn't include comparisons to more recent state-of-the-art UDA techniques. This makes it harder to fully assess the relative advantages of DALUPI in the broader context of current UDA research.

---

> ### Author Response · Authors · 2024-07-05
> **Response to Reviewer jGW2**
>
> We would like to thank Reviewer jGW2 for their insightful comments and questions. We respond to each point in turn below.
>
> **R: The theoretical guarantees of DALUPI rely on the assumption that the privileged information is "sufficient" for predicting labels. While this assumption is justified for some of the experimental tasks, it may not hold in many real-world scenarios. The paper could benefit from a more thorough discussion of when this assumption is likely to hold and how violations of it might impact performance.**
>
> A: We thank the reviewer for pointing this out. In principle, there should exist some PI such that it is sufficient for a given task. However, if sufficiency does not hold then we would expect the overall performance to decrease somewhat while generalization error would be stable. That is, of course, if covariate shift w.r.t. $w$ still holds. We have added some discussion in Section 2.1 regarding this.
>
> **R: While the paper compares DALUPI to some baseline UDA methods (DANN and MDD), it doesn't include comparisons to more recent state-of-the-art UDA techniques. This makes it harder to fully assess the relative advantages of DALUPI in the broader context of current UDA research.**
>
> A: We argue that our chosen baselines and tasks are sufficient for our claims: to give theoretical and empirical evidence for the benefit of learning using privileged information in UDA compared to learning without it (see Section 4.0 paragraph §1--2). SL-S is simple but a strong baseline: On large benchmarks datasets DomainBed and WILDS, there is still no method that \emph{consistently} outperforms SL-S (ERM) without transfer learning (unfortunately, these benchmarks lack privileged information and were therefore not included in this work). We also include methods with theoretical guarantees (DANN, MDD) and a method that makes use of PI (In-N-Out). In all experiments, we show that our approach achieves superior or similar performance to these baselines.
>
>
>
> **R: The authors could design experiments to test how sensitive DALUPI is to violations of the sufficiency assumption. This could involve artificially degrading the quality of the privileged information to see how it affects performance. Such experiments would help understand the robustness of DALUPI in scenarios where the privileged information is not perfectly sufficient for predicting labels.**
>
> A: As we note in, e.g., Section 4.4, some of the experiments which we conduct in the paper may not satisfy the sufficiency assumption. We do agree that it would be interesting to do a deeper investigation into the stability of DALUPI under perturbation of PI. However, we argue that this type of investigation would be extensive and not fit within the scope of our work. We acknowledge that this is an interesting direction for future work in the discussion in Section 6.
>
>
> **R: Another thing to do would be to test DALUPI on more challenging cross-domain adaptation tasks, such as adapting between drastically different image styles (e.g., photos to sketches, or Yelp reviews to tweets). This would showcase the method's ability to handle more extreme domain shifts.**
>
> A: We thank the reviewer for the suggestion and agree that this would be interesting. We have added a short acknowledgement of this direction in the discussion. However, a common issue is that it is often unclear which metric to use to measure what constitutes a larger shift in distribution. Adapting between, to us, radically different image styles or similar is often used as a proxy measure. However, it is true that the method we propose would likely be useful in tackling more radical domain shifts with a good choice of PI that is consistent across domains. For images this would be if you choose PI such that there is limited image style information contained within, e.g. binary attributes in the CelebA task, compared to the input images themselves (which may well be affected by style differences between domains).
>
>
> **R: There is a whole line of work in NLP that calls this learning with feature feedback that you haven't discussed in related work. Several recent papers have shown how learning with feature feedback can help models have better performance vis-a-vis out of domain. Please include those in related work. It might also be beneficial t9o show your evaluate on those NLP tasks. [1, 2]**
>
> **[1] Kaushik, D., Setlur, A., Hovy, E. H., & Lipton, Z. C. Explaining the Efficacy of Counterfactually Augmented Data. In International Conference on Learning Representations. 2021. [2] Katakkar, A., Yoo, C. H., Wang, W., Lipton, Z. C., & Kaushik, D. Practical Benefits of Feature Feedback Under Distribution Shift. In Proceedings of the Fifth BlackboxNLP Workshop on Analyzing and Interpreting Neural Networks for NLP (pp. 346-355). 2022.**
>
> A: We kindly thank the reviewer for the references. We have included these examples in our section on related work.

---

### Author Response · Authors · 2024-07-05
**Revision submitted**

We have submitted a revision of the paper which has incorporated changes requested by the reviewers. The edits made are marked with blue text. Best, The authors

---

### Author Response · Authors · 2024-08-14
**Re: Recommendation**

Dear reviewers and chairs,

We write to enquire about the recommendation for this paper. We submitted our revision on July 5 and were informed that the recommendations were due no later than four weeks after that, on July 22.

Best,
Authors

---

> ### Comment · Action_Editor_yBFy · 2024-08-14
> **Decision available shortly**
>
> Hi,
>
> You should have access to the decision very soon, all that's left is for the Editors in Chief to validate the decision.
>
> Best regards,
>
> Your Action Editor

---

### Decision · Action_Editor_yBFy · 2024-08-13

**Recommendation:** Accept as is

**Comment:**

The paper meets the bar both in terms of Claims and Evidence and in terms of Audience. I would encourage the authors to address Reviewer hqvf's comments on notation and mathematical rigor to the extent that they can for the camera-ready version.

**Audience:**

All three reviewers agree that the paper meets the Audience bar (Reviewer jGW2: "an interesting problem with a wide audience"; Reviewer hqvf: "the message of the paper is rather interesting and publishing this paper could be beneficial to the ML community, in particular, researchers in the domain adaptation (with guarantees) subfield"; Reviewer PbCi: "contribution is of interest for some part of the audience of TMLR").

**Claims And Evidence:**

All three reviewers agree that the paper meets the Claims and Evidence bar. Reviewer PbCi notes that "the paper is well executed and represents a good technical work". Reviewer hqvf "finds that the paper carefully states and documents [its findings] and further supports them with both theoretical and empirical evidence". They also point out that "the paper may benefit from one more revision because there are sometimes inconsistencies in the notation and minor gaps in the mathematical rigor which should be fixed", but given that "the message of the paper is rather interesting and publishing this paper could be beneficial to the ML community" they feel that the submission's strengths outweigh this particular weakness.